# Experiment Planning with Function Approximation

**Aldo Pacchiano**
Broad Institute & Boston University
apacchia@broadinstitute.org

**Jonathan N. Lee**
Stanford University
jnl@stanford.edu

**Emma Brunskill**
Stanford University
ebrun@cs.stanford.edu

## Abstract

We study the problem of experiment planning with function approximation in contextual bandit problems. In settings where there is a significant overhead to deploying adaptive algorithms—for example, when the execution of the data collection policies is required to be distributed, or a human in the loop is needed to implement these policies—producing in advance a set of policies for data collection is paramount. We study the setting where a large dataset of contexts but not rewards is available and may be used by the learner to design an effective data collection strategy. Although when rewards are linear this problem has been well studied [53], results are still missing for more complex reward models. In this work we propose two experiment planning strategies compatible with function approximation. The first is an eluder planning and sampling procedure that can recover optimality guarantees depending on the eluder dimension [42] of the reward function class. For the second, we show that a uniform sampler achieves competitive optimality rates in the setting where the number of actions is small. We finalize our results introducing a statistical gap fleshing out the fundamental differences between planning and adaptive learning and provide results for planning with model selection.

## 1 Introduction

Data-driven decision-making algorithms have achieved impressive empirical success in various domains such as online personalization [4, 48], games [36, 43], dialogue systems [32] and robotics [25, 35]. In many of these decision-making scenarios, it is often advantageous to consider contextual information when making decisions. This recognition has sparked a growing interest in studying adaptive learning algorithms in the setting of contextual bandits [26, 33, 6] and reinforcement learning (RL) [46]. Adaptive learning scenarios involve the deployment of data collection policies, where learners observe rewards or environment information and utilize this knowledge to shape subsequent data collection strategies. Nonetheless, the practical implementation of adaptive policies in real-world experiments currently presents significant challenges. First, there is significant infrastructure requirements and associated overheads which require skills and resources many organizations lack. For example, while there are end-user services that enable organizations to automatically send different text messages to different individuals, such services typically do not offer adaptive bandit algorithms. Second, the resulting reward signal may be significantly delayed. As an example, the effect of personalized health screening reminders on a patient making a doctors appointments may take weeks. Therefore, while it is increasingly recognized that there exist other settings where context-specific policies are likely to be beneficial, including behavioural science[8], many organizations working in these settings may find it infeasible to run experiments with adaptive policies. To bridge this gap, there is a need to explore the deployment of non-adaptive or static strategies that can effectively collect data with minimal or no updates. Surprisingly, limited research has been conducted to investigate this particular setting, highlighting the importance of further exploration in this area.

37th Conference on Neural Information Processing Systems (NeurIPS 2023).

In this work, we consider how to design a static experimental sampling strategy for contextual multi-armed bandit settings which, when executed, will yield a dataset from which we can compute a near-optimal contextual policy (one with small simple regret). Our framework closely adheres to the experiment planning problem formulation originally introduced by [53]. The problem involves a learner interacting with a contextual bandit problem where the reward function is unknown. Our assumption is that the learner possesses a substantial offline dataset comprising of $m$ sample contexts, none of which include reward information. The learner's objective is to utilize this data to design a static policy sequence of length $T$ (where $m \geqslant T$) that enables the collection of valuable reward information when deployed in the real world. The ultimate goal is to produce an almost optimal policy using the data gathered during deployment. To address this scenario, the authors of [53] propose a two-phase approach involving Planner and Sampler algorithms. In the planner phase, the learner employs the context data to generate a set of sampling policies, which are then utilized in the sampler phase. The primary focus of [53] lies in the analysis of the linear case, assuming the reward function to be a linear function of the context vectors. Their key algorithmic contribution is the linear Planner Algorithm, which encompasses a reward-free instance of $\mathrm{LinUCB}$. The analysis presented in [53] demonstrates that the required number of samples under the static sampler to achieve an $\varepsilon$-optimal policy (also referred to as simple regret) scales as $\mathcal{O}\left(d^2/\varepsilon^2\right)$, where $d$ represents the dimension of the underlying space. This result matches the online minimax sample complexity for the simple regret problem [11, 1].

The algorithm proposed by [53] effectively constructs a static policy sequence of adequate span, utilizing the linear structure of the problem. However, in many problem settings, linearity alone may not be sufficient to capture the true nature of the underlying reward model. This limitation is particularly evident in scenarios such as genetic perturbation experimentation [40] or other similar contexts. In such cases, the availability of algorithms that do not rely on linearity becomes crucial. Unfortunately, extending the results and techniques presented in [53] to the generic function approximation regime is not straightforward.

Here we consider when the reward function is realized by an unknown function $f_\star$ belonging to a known function class $\mathcal{F}$ (which can be more complex than linear).

**Adaptive Learning with Function Approximation.** In adaptive learning settings where a learner can change the data collection policies as it observes rewards and has much more flexibility than in experiment planning scenarios, various adaptive learning procedures compatible with generic function approximation have been proposed for contextual bandit problems. Among these, two significant methods relevant to our discussion are the Optimistic Least Squares algorithm ($\mathrm{OptLS}$) introduced by [42] and the SquareCB algorithm introduced by [17]. Both of these methods offer guarantees for cumulative regret. Specifically, the cumulative regret of $\mathrm{OptLS}$ scales as $\mathcal{O}(\sqrt{d_{\mathrm{eluder}} \log(|\mathcal{F}|)T})$, while the cumulative regret of $\mathrm{SquareCB}$ scales as $\mathcal{O}\left(\sqrt{|\mathcal{A}| \log(|\mathcal{F}|)T}\right)$, where $\mathcal{A}$ corresponds to the set of actions. The eluder dimension[1] ($d_{\mathrm{eluder}}$) is a statistical complexity measure introduced by [42], which enables deriving guarantees for *adaptive* learning algorithms based on the principle of optimism in the face of uncertainty in contextual bandits and reinforcement learning [31, 22, 37, 9]. By employing an online-to-batch conversion strategy, these two methods imply that the number of samples required to achieve $\varepsilon$-simple regret using an adaptive algorithm is at most $\mathcal{O}\left(d_{\mathrm{eluder}} \log(|\mathcal{F}|)/\varepsilon^2\right)$ and $\mathcal{O}\left(|\mathcal{A}| \log(|\mathcal{F}|)/\varepsilon^2\right)$ respectively.

**Contributions.** In this paper, we address the function approximation setting within the contextual bandit (static) experiment planning problem. Although in experiment planning the data collection policies have to be produced before interacting with the environment, we establish that surprisingly the adaptive $\varepsilon$-simple regret rates achieved by $\mathrm{OptLS}$ and $\mathrm{SquareCB}$ are also attainable by a static experiment sampling strategy. To achieve this, we introduce and analyze two experimental planning algorithms. The $\mathrm{PlannerEluder}$ algorithm (see Section 4) utilizes confidence intervals derived from least squares optimization to construct a static set of policies to be executed during the sampling phase. When the algorithm has access to a sufficient number of offline samples to generate $T = \Omega\left(d_{\mathrm{eluder}} \log(|\mathcal{F}|)/\varepsilon^2\right)$ static policies, the learner can produce an $\varepsilon$-optimal policy denoted as $\widehat{\pi}_T$ thus matching the online-to-batch $\varepsilon$-simple regret rates of $\mathrm{OptLS}$. Since the eluder dimension of linear function classes of dimension $d$ is -up to logarithmic factors- of order $\mathcal{O}(d)$, our results

---

[1]We formally introduce this quantity in Section 4. Here we use a simpler notation to avoid confusion.

recover the linear experiment planning rates of [53]. Additionally, in Section 5, we demonstrate that collecting $T = \Omega\left(|\mathcal{A}|\log(|\mathcal{F}|)/\varepsilon^2\right)$ uniform samples is sufficient to obtain an $\varepsilon$-optimal policy $\widehat{\pi}_T$. This matches the online-to-batch $\varepsilon$-simple regret rates of SquareCB . These two results yield conclusions similar to those known in the linear setting [53], but for general function classes.

These results prompt two important questions: (1) are existing adaptive cumulative regret algorithms already optimal for simple regret minimization, and (2) is static experiment planning sufficient to match the simple regret guarantees of adaptive learning for realizable contextual bandits?

In Section 6, we provide a negative answer to both questions. We present a certain structured class of bandit problems where a different adaptive algorithm can require significantly less samples than that implied by the bounds from OptLS and SquareCB . Our result is for the *realizable* setting [2, 18, 17, 44, 9], where the true reward function $f_\star$ belongs to a known function class $\mathcal{F}$, where the true complexity of simple regret problems in adaptive learning scenarios has not be known known in this setting. This result complements recent results [34] that online-to-batch strategies are suboptimal for minimizing simple regret in the agnostic contextual bandits setting.[2]

For the second question, our statistical lower bound shows that for this class of problems, a significantly smaller number of samples are needed to find an $\varepsilon$-optimal policy when using adaptive learning, than when using a static policy. This result complements related work in reinforcement learning with realizable linear state-action value function approximation[52].

Finally, we address the problem of model selection when the learner is presented with a family of reward function classes $\{\mathcal{F}_i\}_{i=1}^M$ and is guaranteed that the true reward function $f_\star$ belongs to $\mathcal{F}_{i_\star}$, where the index $i_\star$ is unknown. We demonstrate that as long as $T = \Omega\left(|\mathcal{A}|\log(\max(M,|\mathcal{F}_{i_\star}|))/\varepsilon^2\right)$ uniform samples are available, it is possible to construct an $\varepsilon$-optimal policy denoted as $\widehat{\pi}_T$. Further details regarding these results can be found in Section 7.

## 2 Further Related Work

**Best Arm Identification and Design of Experiments.** Previous work to efficiently produce an almost optimal policy for non-contextual linear and multi-armed bandits settings is based on best-arm identification procedures [16, 20, 13, 45, 47, 51]. These are strategies that react adaptively to the collected rewards and achieve instance dependent sample complexity. Other papers in the design of experiments literature [24, 15, 29, 41] introduce methods for designing a non-adaptive policy that can be used to find an optimal arm with high probability in non-contextual scenarios.

**Safe Optimal Design.** When safety concerns are imposed in the exploration policy, works such as [54] and [49] have studied the problem of producing a safe and efficient static policy used to collect data that can then be used for policy evaluation in the MAB and linear settings. This is in contrast with the unconstrained contextual function approximation scenario we study.

**Reward Free Reinforcement Learning.** In Reinforcement Learning settings a related line of research [21, 50, 10] considers the problem of producing enough exploratory data to allow for policy optimization for all functions in a reward class. These works allow for the learner to interact with the world for a number of steps, enough to collect data that will allow for zero-shot estimation of the optimal policy within a reward family. This stands in contrast with the experiment planning setup, where the objective is to produce a sequence of policies for static deployment at test time with the objective of learning the unknown reward function.

## 3 Problem Definition

We study a two-phase interaction between a learner and its environment consisting of a *Planning* and a *Sampling* phase. During the planning phase the learner has access to $m \geqslant T$ i.i.d. offline

---

[2]In the agnostic case the learner is instead presented with an arbitrary policy class $\Pi$ that may or may not be related to the mean reward function $f_\star$. In this setting a recent study by [34] introduces an algorithm that achieves sharp rates characterized in terms of a quantity $\rho_\Pi$.

context samples[3] from a context distribution $\mathcal{P}$ supported on a context space $\mathcal{X}$ as well as knowledge of a reward function class $\mathcal{F}$ where each element $f \in \mathcal{F}$ has domain $\mathcal{X} \times \mathcal{A}$ where $\mathcal{A}$ is the action set. During the sampling phase the learner interacts with a contextual bandit problem for $T$ steps where at each time-step $t$ a context $x_t \sim \mathcal{P}$ is revealed to the learner, the learner takes an action $a_t$ and receives a reward $r_t$. We adopt the commonly used realizability assumption, which is widely employed in the contextual bandit literature [2, 18, 17, 44, 9].

**Assumption 3.1** (Realizability). *There exists a function $f_\star \in \mathcal{F}$ such that $r_t = f_\star(x_t, a_t) + \xi_t$ where $f_\star \in \mathcal{F}$ and $\xi_t$ is a conditionally zero mean $1-$subgaussian random variable for all $t \in [T]$ during the sampling phase.*

During the planning phase the learner is required to build a static policy sequence $\{\pi_i\}_{i=1}^{T}$ (a policy is a mapping from $\mathcal{X}$ to $\Delta_{\mathcal{A}}$, the set of distributions over actions) that, without knowledge of the reward, can be deployed to collect data during the sampling phase. In contrast with adaptive learning procedures, in the experiment planning problem the policies in $\{\pi_i\}_{i=1}^{T}$ have to be fully specified during the planning phase and cannot depend on the learner's observed reward values during the sampling phase. The learner's objective is to use the reward data collected during the sampling phase to produce an $\varepsilon-$optimal policy $\widehat{\pi}_T$ such that,

$$\mathbb{E}_{x \sim \mathcal{P}, a \sim \widehat{\pi}_T(x)} [f_\star(x, a)] + \varepsilon \geq \max_{\pi} \mathbb{E}_{x \sim \mathcal{P}, a \sim \pi(x)} [f_\star(x, a)].$$

for a suboptimality parameter $\varepsilon > 0$. Because we have assumed realizability of the reward function, the optimal policy among all possible policies $\pi_\star = \arg\max_{\pi} \mathbb{E}_{x \sim \mathcal{P}, a \sim \pi(x)} [f_\star(x, a)]$ can be written as $\pi_\star(x) = \arg\max_{a \in \mathcal{A}} f_\star(x, a)$ for all $x \in \mathcal{X}$.

Throughout this work we make the following boundedness assumptions,

**Assumption 3.2** (Bounded Function Range). *The range of $\mathcal{F}$ is bounded, i.e. for all $x, a \in \mathcal{X} \times \mathcal{A}$, $|f(x, a)| \leq B$ for some constant $B > 0$.*

**Assumption 3.3** (Bounded Noise). *The noise variables are bounded $|\xi_t| \leq \bar{B}$ for all $t \in [T]$ for some constant $\bar{B} > 0$.*

Assumptions 3.2 and 3.3 imply $|r_t| \leq B + \bar{B}$ for all $t \in [T]$.

# 4 Planning Under the Eluder Dimension

The eluder dimension is a sequential notion of complexity for function classes that was originally introduced by [42]. The eluder dimension can be used to bound the regret of optimistic least squares algorithms in contextual bandits [42, 9] and reinforcement learning [37]. Informally speaking the eluder dimension is the length of the longest sequence of points one must observe to accurately estimate the function value at any other point. We use the eluder dimension definition from [42].

**Definition 4.1.** *($\varepsilon-$dependence) Let $\mathcal{G}$ be a scalar function class with domain $\mathcal{Z}$ and $\varepsilon > 0$. An element $z \in \mathcal{Z}$ is $\varepsilon-$dependent on $\{z_1, \cdots, z_n\} \subseteq \mathcal{Z}$ w.r.t. $\mathcal{G}$ if any pair of functions $g, g' \in \mathcal{G}$ satisfying $\sqrt{\sum_{i=1}^{n}(g(z_i) - g'(z_i))^2} \leq \varepsilon$ also satisfies $g(z) - g'(z) \leq \varepsilon$. Furthermore, $z \in \mathcal{Z}$ is $\varepsilon-$independent of $\{z_1, \cdots, z_n\}$ w.r.t. $\mathcal{G}$ if it is not $\varepsilon-$dependent on $\{z_1, \cdots, z_n\}$.*

**Definition 4.2.** *($\varepsilon$-eluder) The $\varepsilon-$non monotone eluder dimension $\overline{d_{\mathrm{eluder}}}(\mathcal{G}, \varepsilon)$ of $\mathcal{G}$ is the length of the longest sequence of elements in $\mathcal{Z}$ such that every element is $\varepsilon-$independent of its predecessors. Moreover, we define the $\varepsilon-$eluder dimension $d_{\mathrm{eluder}}(\mathcal{G}, \varepsilon)$ as $d_{\mathrm{eluder}}(\mathcal{G}, \varepsilon) = \max_{\varepsilon' \geq \varepsilon} \overline{d_{\mathrm{eluder}}}(\mathcal{G}, \varepsilon)$.*

By definition the $\varepsilon-$eluder dimension increases as $\varepsilon$ is driven down to zero. Let $\mathcal{D}$ be a dataset of $\mathcal{X}$, $\mathcal{A}$ pairs. For any $f, f'$ with support $\mathcal{X} \times \mathcal{A}$ we use the notation $\|f - f'\|_{\mathcal{D}}$ to denote the data norm of the difference between functions $f$ and $f'$:

$$\|f - f'\|_{\mathcal{D}} = \sqrt{\sum_{(x,a) \in \mathcal{D}} (f(x, a) - f'(x, a))^2}.$$

---

[3]This assumption is based on the fact that gathering offline context samples can be significantly less costly compared to conducting multiple rounds of experimental deployment.

---

**Algorithm 1** EluderPlanner

1: Input: $m \geqslant T$ samples $\{x_\ell \sim \mathcal{P}\}_{\ell=1}^m$, function class $\mathcal{F}$, confidence radius function $\beta_{\mathcal{F}} : t, \delta \to \mathbb{R}$.
2: Initialize data buffer $\mathcal{D}_1 = \varnothing$
3: **for** $t = 1 \ldots T$ **do**
4:     For any $x \in \mathcal{X}$ define the uncertainty radius of action $a \in \mathcal{A}$ as,

$$\omega(x, a, \mathcal{D}_t) = \max_{f, f' \in \mathcal{F} \text{ s.t. } \|f - f'\|_{\mathcal{D}_t} \leqslant 4\beta_{\mathcal{F}}(t, \delta)} f(x, a) - f'(x, a).$$

5:     Define the policy $\pi_t$ as $\pi_t(x) = \arg\max_{a \in \mathcal{A}} \omega(x, a, \mathcal{D}_t)$ for all $x \in \mathcal{X}$.
6:     Update $\mathcal{D}_{t+1} \leftarrow \mathcal{D}_t \cup \{(x_t, \pi_t(x_t))\}$
7: **end for**

---

---

**Algorithm 2** Sampler

1: Input: number of time-steps $T$.
2: Initialize data buffer $\widetilde{\mathcal{D}}_1 = \varnothing$
3: **for** $t = 1 \ldots T$ **do**
4:     Define deployment policy $\tilde{\pi}_t$ as $\pi_t$.
5:     Observe context $\tilde{x}_t \sim \mathcal{P}$.
6:     Play action $\tilde{a}_t = \tilde{\pi}_t(\tilde{x}_t)$ and receive reward $\tilde{r}_t$.
7:     Update $\widetilde{\mathcal{D}}_{t+1} \leftarrow \widetilde{\mathcal{D}}_t \cup \{(\tilde{x}_t, \tilde{a}_t, \tilde{r}_t)\}$.
8: **end for**

---

The EluderPlanner algorithm takes as input $m \geqslant T$ i.i.d. samples from the context distribution $\{x_\ell \sim \mathcal{P}\}_{\ell=1}^m$ and a realizable function class $\mathcal{F}$ satisfying Assumption 3.1. The learner iterates over $T$ out of these $m$ samples in a sequential manner. We use the name $x_t$ to denote the $t-$th input sample and call $\pi_t$ to the (deterministic) exploration policies produced upon processing samples $1, \cdots, t-1$. We use the notation $\mathcal{D}_t = \{(x_\ell, \pi_\ell(x_\ell))\}_{\ell=1}^{t-1}$ to denote the dataset composed of the first $t-1$ (ordered) samples from $\{x_\ell\}_{\ell=1}^m$ and actions from policies $\{\pi_\ell\}_{\ell=1}^{t-1}$. We adopt the convention that $\mathcal{D}_1 = \varnothing$. The output of the EluderPlanner algorithm is the sequence of policies $\{\pi_t\}_{t=1}^T$. Policy $\pi_t$ is defined as $\pi_t(x) = \arg\max_{a \in \mathcal{A}} \omega(x, a, \mathcal{D}_t)$ such that $\omega(x, a, \mathcal{D}_t)$ is an uncertainty measure defined as,

$$\omega(x, a, \mathcal{D}_t) = \max_{f, f' \in \mathcal{F} \text{ s.t. } \|f - f'\|_{\mathcal{D}_t} \leqslant 4\beta_{\mathcal{F}}(t, \delta)} f(x, a) - f'(x, a). \tag{1}$$

Where the confidence radius function $\beta_{\mathcal{F}}(\delta, t)$ equals

$$\beta_{\mathcal{F}}(\delta, t) = \bar{C}(B + \bar{B})\sqrt{\log\left(\frac{|\mathcal{F}|t}{\delta}\right)} \tag{2}$$

for some universal constant $\bar{C} > 0$ defined in Proposition A.7. This is a complexity radius implied by the guarantees of least squares regression (Lemma A.8) and constraints imposed by Lemma 4.6. See Appendix A.1 and Proposition A.7 for a detailed discussion about this definition. We call the combination of the EluderPlanner and Sampler as the *eluder planning algorithm*. Its performance guarantees are obtained by relating the regret of a constructed sequence of optimistic policies $\{\widetilde{\pi}_t^{\text{opt}}\}_{t=1}^T$ based on the data collected by $\{\pi_t\}_{t=1}^T$ to the sum of uncertainty radii $\sum_{t=1}^T \omega(x_t, \pi_t(x_t), \mathcal{D}_t)$ of the context-actions collected during the planning phase. Using optimism when constructing this policy sequence is important. In contrast with the sequence of greedy policies $\pi_t^{\text{greedy}}(\cdot|x) = \arg\max_{a \in \mathcal{A}} \widehat{f}_t(x, a)$, where $\widehat{f}_t = \arg\min_{f \in \mathcal{F}} \sum_{(x,a,r) \in \widetilde{\mathcal{D}}_t} (f(x, a) - r)^2$ resulting from a least squares over $\widetilde{\mathcal{D}}_t$, the sequence of optimistic policies satisfies a regret bound. Thus, playing a uniform policy from the sequence $\{\widetilde{\pi}_t^{\text{opt}}\}_{t=1}^T$ satisfies a simple regret guarantee. The sum of uncertainty radii $\sum_{t=1}^T \omega(x_t, \pi_t(x_t), \mathcal{D}_t)$ is bounded using a Lemma we borrow from [9].

**Lemma 4.3.** *[Lemma 3 from [9]] Let $\mathcal{F}$ be a function class satisfying Assumption 3.2 with $\varepsilon$-eluder dimension $d_{\text{eluder}}(\mathcal{F}, \varepsilon)$. For all $T \in \mathbb{N}$ and any dataset sequence $\{\bar{D}_t\}_{t=1}^\infty$ with $\bar{D}_1 = \varnothing$ and $\bar{D}_t = \{\{\bar{x}_\ell, \bar{a}_\ell\}\}_{\ell=1}^{t-1}$ of context-action pairs, the following inequality on the sum of the uncertainty radii holds,*

$$\sum_{t=1}^T \omega(\bar{x}_t, \bar{a}_t, \bar{D}_t) \leqslant \mathcal{O}\left(\min\left(BT, Bd_{\text{eluder}}(\mathcal{F}, B/T) + \beta_{\mathcal{F}}(T, \delta)\sqrt{d_{\text{eluder}}(\mathcal{F}, B/T)T}\right)\right).$$

**Extracting Policy $\widehat{\pi}_T$ from Data.** At the end of the execution of the Sampler algorithm, the learner will have access to a sequence of datasets $\{\widetilde{D}_t\}_{t=1}^T$ of contexts, actions and reward triplets. The learner will use this sequence of datasets to compute the sequence of regression functions,

$$\widehat{f}_t = \arg\min_{f \in \mathcal{F}} \sum_{\ell=1}^{t-1} (f(\tilde{x}_\ell, \tilde{a}_\ell) - \tilde{r}_\ell)^2. \tag{3}$$

Standard Least Squares (LS) results (see Lemma A.8) imply that $\|\widehat{f}_t - f_\star\|_{\widetilde{\mathcal{D}}_t} \leqslant \beta(t, \delta)$ with high probability for all $t \in \mathbb{N}$. These confidence sets are used to define a sequence of optimistic (deterministic) policies $\{\widetilde{\pi}_t^{\mathrm{opt}}\}_{t=1}^T$:

$$\widetilde{\pi}_t^{\mathrm{opt}}(x) = \arg\max_{a \in \mathcal{A}} \max_{\tilde{f} \text{ s.t. } \|\widehat{f}_t - f\|_{\widetilde{\mathcal{D}}_t} \leqslant \beta_{\mathcal{F}}(t, \delta)} \tilde{f}(x, a) \tag{4}$$

The candidate optimal policy $\widehat{\pi}_T$ is then defined as $\widehat{\pi}_T = \mathrm{Uniform}(\widetilde{\pi}_1^{\mathrm{opt}}, \cdots, \widetilde{\pi}_T^{\mathrm{opt}})$. The main result in this section is the following Theorem.

**Theorem 4.4.** *Let $\varepsilon > 0$. There exists a universal constant $c > 0$ such that if*

$$T \geqslant c \frac{\max^2(B, \bar{B}, 1) d_{\mathrm{eluder}}(\mathcal{F}, B/T) \log\left(|\mathcal{F}|T/\delta\right)}{\varepsilon^2} \tag{5}$$

*then with probability at least $1 - \delta$ the eluder planning algorithm's policy $\widehat{\pi}_T$ is $\varepsilon$-optimal .*

**Remark 4.5.** *The results of this section hold for the structured bandits setting [23, 27]; that is, scenarios where $\mathcal{X} = \{\varnothing\}$.*

Both sides of the inequality defining $T$ in Theorem 4.4 depend on $T$. Provided $d_{\mathrm{eluder}}(\mathcal{F}, B/T)$ is sublinear as a function of $T$, there exists a finite value of $T$ satisfying Equation 5. When $d_{\mathrm{eluder}}(\mathcal{F}, \bar{\varepsilon}) = \bar{d}_{\mathrm{eluder}} \log\left(1/\bar{\varepsilon}\right)$ for some $\bar{d}_{\mathrm{eluder}}$ and for all $\bar{\varepsilon} > 0$ setting $T \geqslant c \frac{\max^2(B, \bar{B}, 1) d_{\mathrm{eluder}}(\mathcal{F}, B\varepsilon) \log(|\mathcal{F}|/\varepsilon\delta)}{\varepsilon^2}$ is enough to guarantee the conditions of Theorem 4.4 are satisfied. Examples 4 and 5 from [42] show the eluder dimension of linear or generalized linear function classes can be written in such way. In these cases $\bar{d}_{\mathrm{eluder}}$ is a constant multiple of the underlying dimension of the space.

Using standard techniques, the results of Theorem 4.4 can be adapted to the case when the function class $\mathcal{F}$ is infinite but admits a metrization and has a covering number. In this case the sample complexity will scale not with $\log(|\mathcal{F}|)$ but instead with the logarithm of the covering number of $\mathcal{F}$. For example, in the case of linear functions, the logarithm of the covering number of $\mathcal{F}$ under the $\ell_2$ norm will scale as $d$, the ambient dimension of the space up to logarithmic factors of $T$ (see for example Lemma D.1 of [14]). Plugging this into the sample guarantees of Theorem 4.4 recovers the $\widetilde{\mathcal{O}}\left(d^2/\varepsilon^2\right)$ sample complexity for linear experiment planning from [53].

## 4.1 Proof Sketch of Theorem 4.4

Let $\tilde{x}_1, \cdots, \tilde{x}_T$ be the sampler's context sequence. The proof works by showing that with probability at least $1 - \delta$ the regret of the $\{\widetilde{\pi}_t^{\mathrm{opt}}(\cdot)\}_{t=1}^T$ sequence satisfies the regret bound

$$\sum_{t=1}^T \max_{a \in \mathcal{A}} f_\star(\widetilde{x}_t, a) - f_\star(\widetilde{x}_t, \widetilde{\pi}_t^{\mathrm{opt}}(\widetilde{x}_t)) \leqslant \mathcal{O}\left((B + \bar{B})\sqrt{d_{\mathrm{eluder}}(\mathcal{F}, B/T)T \log(|\mathcal{F}|T/\delta)}\right)$$

A key part of the proof involves relating the balls defined by the data norms of the planner datasets $\{\mathcal{D}_t\}_{t=1}^T$ and those induced by the data norms of the sampler datasets $\{\widetilde{\mathcal{D}}_t\}_{t=1}^T$. The precise statement of this relationship is provided in Lemma 4.6, and its proof can be found in Appendix C.

**Lemma 4.6.** *With probability at least $1 - \frac{\delta}{12}$,*

$$\{(f, f') \text{ s.t. } \|f - f'\|_{\widetilde{\mathcal{D}}_t} \leqslant 2\beta_{\mathcal{F}}(t, \delta)\} \subseteq \{(f, f') \text{ s.t. } \|f - f'\|_{\mathcal{D}_t} \leqslant 4\beta_{\mathcal{F}}(t, \delta)\} \tag{6}$$

*Simultaneously for all $t \in [T]$. Where $\{D_t\}_{t=1}^T$ is the dataset sequence resulting of the execution of* EluderPlanner *while $\{\widetilde{D}_t\}_{t=1}^T$ is the dataset sequence resulting of the execution of the* Sampler *and $\beta_{\mathcal{F}}(\delta, t)$ is the confidence radius function defined in Equation 2.*

Let $\widehat{f}_t$ as in Equation 3 and define $\mathcal{E}$ as the event where the Standard Least Squares results (see Lemma A.8) hold i.e. $\|\widehat{f}_t - f_\star\|_{\widetilde{\mathcal{D}}_t} \leqslant \beta_{\mathcal{F}}(t, \delta)$ for all $t \in [T]$ and also Equation 6 from Lemma 4.6 holds for all $t \in [T]$. The results of Lemmas A.8 and 4.6 imply $\mathbb{P}(\mathcal{E}) \geqslant 1 - \frac{\delta}{6}$.

For context $x$ and action $a$ let's denote by $\tilde{f}_t^{x,a}$ as a function achieving the inner maximum in the definition of $\tilde{\pi}_t^{\mathrm{opt}}$ (see Equation 4). When $\mathcal{E}$ holds the policies $\{\tilde{\pi}_t^{\mathrm{opt}}\}_{t=1}^T$ are optimistic in the sense that $\tilde{f}_t^{x,\tilde{\pi}_t^{\mathrm{opt}}(x)}(x, \tilde{\pi}_t^{\mathrm{opt}}(x)) \geqslant \max_{a \in \mathcal{A}} f_\star(x, a)$ and therefore,

$$
\sum_{t=1}^T \max_{a \in \mathcal{A}} f_\star(\tilde{x}_t, a) - f_\star(\tilde{x}_t, \tilde{\pi}_t^{\mathrm{opt}}(\tilde{x}_t)) \stackrel{(i)}{\leqslant} \sum_{t=1}^T \tilde{f}_t^{x,\tilde{\pi}_t^{\mathrm{opt}}(x)}(\tilde{x}_t, \tilde{\pi}_t^{\mathrm{opt}}(\tilde{x}_t)) - f_\star(\tilde{x}_t, \tilde{\pi}_t^{\mathrm{opt}}(\tilde{x}_t))
$$

$$
\stackrel{(ii)}{\leqslant} \sum_{t=1}^T \max_{f,f' \in \mathbb{B}_t(2, \widetilde{\mathcal{D}}_t)} f(\tilde{x}_t, \tilde{\pi}_t^{\mathrm{opt}}(\tilde{x}_t)) - f'(\tilde{x}_t, \tilde{\pi}_t^{\mathrm{opt}}(\tilde{x}_t))
$$

$$
\stackrel{(iii)}{\leqslant} \sum_{t=1}^T \max_{f,f' \in \mathbb{B}_t(4, \mathcal{D}_t)} f(\tilde{x}_t, \tilde{\pi}_t^{\mathrm{opt}}(\tilde{x}_t)) - f'(\tilde{x}_t, \tilde{\pi}_t^{\mathrm{opt}}(\tilde{x}_t)) = (\star).
$$

Where $\mathbb{B}_t(\gamma, \mathcal{D}) = \{f, f' \in \mathcal{F} \text{ s.t. } \|f - f'\|_{\mathcal{D}} \leqslant \gamma \beta_{\mathcal{F}}(\delta, t)\}$. Inequality $(i)$ follows by optimism, $(ii)$ is a consequence of $\tilde{f}_t^{x,\tilde{\pi}_t^{\mathrm{opt}}(x)}, f_\star \in \mathbb{B}_t(2, \widetilde{\mathcal{D}}_t)$ when $\mathcal{E}$ holds since in this case $\|\widehat{f}_t - f_\star\|_{\widetilde{\mathcal{D}}_t} \leqslant \beta_{\mathcal{F}}(t, \delta)$ and $\|\tilde{f}_t^{x,\tilde{\pi}_t^{\mathrm{opt}}(x)} - \widehat{f}_t\|_{\widetilde{\mathcal{D}}_t} \leqslant \beta_{\mathcal{F}}(t, \delta)$, and $(iii)$ follows because when $\mathcal{E}$ holds, Equation 6 of Lemma 4.6 is satisfied and therefore $\mathbb{B}_t(2, \widetilde{\mathcal{D}}_t) \subseteq \mathbb{B}_t(4, \mathcal{D}_t)$. The RHS $(\star)$ of the equation above satisfies,
$$
(\star) = \sum_{t=1}^T \omega(\tilde{x}_t, \tilde{\pi}_t^{\mathrm{opt}}(\tilde{x}_t), \mathcal{D}_t) \leqslant \sum_{t=1}^T \max_{a \in \mathcal{A}} \omega(\tilde{x}_t, a, \mathcal{D}_t) = \sum_{t=1}^T \omega(\tilde{x}_t, \pi_t(\tilde{x}_t), \mathcal{D}_t).
$$

We relate the sum of uncertainty radii $\{\omega(\tilde{x}_t, \pi_t(\tilde{x}_t), \mathcal{D}_t)\}_{t=1}^T$ with those of the planner $\{\omega(x_t, \pi_t(x_t), \mathcal{D}_t)\}_{t=1}^T$ via Hoeffding Inequality (see Lemma A.1) and conclude that w.h.p,

$$
\sum_{t=1}^T \max_{a \in \mathcal{A}} f_\star(\tilde{x}_t, a) - f_\star(\tilde{x}_t, \tilde{\pi}_t^{\mathrm{opt}}(\tilde{x}_t)) \leqslant \sum_{t=1}^T \omega(x_t, \pi_t(x_t), \mathcal{D}_t) + \mathcal{O}\left(B\sqrt{T \log(1/\delta)}\right).
$$

Lemma 4.3 allows us to bound the sum of these uncertainty radii as

$$
\sum_{t=1}^T \omega(x_t, \pi_t(x_t), \mathcal{D}_t) \leqslant \mathcal{O}\left(\min\left(BT, Bd_{\mathrm{eluder}}(\mathcal{F}, B/T) + \beta_{\mathcal{F}}(T, \delta)\sqrt{d_{\mathrm{eluder}}(\mathcal{F}, B/T)T}\right)\right),
$$

and therefore w.h.p,

$$
\sum_{\ell=1}^T \max_{a \in \mathcal{A}} f_\star(\tilde{x}_\ell, a) - f_\star(\tilde{x}_\ell, \tilde{\pi}_j^{\mathrm{opt}}(\tilde{x}_\ell)) \leqslant \mathcal{O}\left(\min\left(BT, Bd_{\mathrm{eluder}}(\mathcal{F}, B/T) + \beta_{\mathcal{F}}(T, \delta)\sqrt{d_{\mathrm{eluder}}(\mathcal{F}, B/T)T}\right)\right).
$$

Converting this cumulative regret bound into a simple regret one (Lemma B.1) finalizes the result. A detailed version of the proof can be found in Appendix C.1.

# 5 Uniform Sampling Strategies

In this section we show that a uniform sampling strategy can produce an $\varepsilon$ optimal policy with probability at least $1 - \delta$ after collecting $\mathcal{O}\left(\frac{\max^2(B, \bar{B})|\mathcal{A}| \log(|\mathcal{F}|/\varepsilon\delta)}{\varepsilon^2}\right)$ samples. This procedure achieves the same simple regret rate as converting SquareCB 's cumulative regret into simple regret[4].

In contrast with the eluder planning algorithm the uniform sampling strategy does not require a planning phase. Instead it consists of running of the Sampler (Algorithm 2) setting $\tilde{\pi}_j = \mathrm{Uniform}(\mathcal{A})$ for all $j \in [T]$. Given a dataset $\widetilde{\mathcal{D}}_T$ of contexts, actions and rewards collected during the sampling phase, we solve the least squares problem:

$$
\widehat{f}_T = \arg\min_{f \in \mathcal{F}} \sum_{(x_i, a_i, r_i) \in \widetilde{D}_T} (f(x_i, a_i) - r_i)^2
$$

and define the policy $\widehat{\pi}_T$ as $\widehat{\pi}_T(x) = \arg\max_{a \in \mathcal{A}} \widehat{f}_T(x, a)$. The main result of this section is,

---

[4]The regret bound of the SquareCB algorithm scales as $\mathcal{O}\left(\sqrt{|\mathcal{A}| \log(\mathcal{F})T \log(T/\delta)}\right)$.

**Theorem 5.1.** *There exists a universal constant $\tilde{c} > 0$ such that if $T \geqslant \tilde{c} \frac{\max^2(B,\bar{B})|\mathcal{A}|\log(|\mathcal{F}|/\varepsilon\delta)}{\varepsilon^2}$ then with probability at least $1 - \delta$ the uniform planning algorithm's policy $\hat{\pi}_T$ is $\varepsilon-$optimal.*

The proof of Theorem 5.1 can be found in Appendix D.

**Comparison Between** OptLS **and** SquareCB **.** The regret rate after $T$ steps of the OptLS algorithm applied to a contextual bandit problem with discrete function class $\mathcal{F}$ scales[5] (up to logarithmic factors) as $\mathcal{O}\left(\sqrt{d_{\mathrm{eluder}}(\mathcal{F}, B/T)\log(|\mathcal{F}|/\delta)T}\right)$. In contrast, the regret of the SquareCB algorithm satisfies a regret guarantee (up to logarithmic factors) of order $\mathcal{O}\left(\sqrt{|\mathcal{A}|\log(|\mathcal{F}|/\delta)T}\right)$ where the eluder dimension dependence is substituted by a polynomial dependence on the number of actions $|\mathcal{A}|$. When the number of actions is small, or even constant, the regret rate of SquareCB can be much smaller than that of OptLS . The opposite is true when the number of actions is large or even infinite. Converting these cumulative regret bounds to simple regret implies the number of samples required to produce an $\varepsilon-$optimal policy from the *adaptive* OptLS policy sequence scales as $\frac{d_{\mathrm{eluder}}(\mathcal{F}, B/T)\log(|\mathcal{F}|/\delta)}{\varepsilon^2}$ whereas for the *adaptive* SquareCB policy sequence it scales as $\frac{|\mathcal{A}|\log(|\mathcal{F}|/\delta)}{\varepsilon^2}$. The results of Theorems 4.4 and 5.1 recover these rates in the experiment planning setting.

# 6 Gap Between Experiment Planning and Adaptive Learning

The results of Sections 4 and 5 imply planning bounds that are comparable to the corresponding online-to-batch guarantees for OptLS [42] and SquareCB [17]. The main result of this section Theorem 6.1 shows there are problems where the number of samples required for experiment planning can be substantially larger than the number of samples required of an adaptive learning algorithm. This result implies the suboptimality of algorithms such as SquareCB and OptLS for adaptive learning.

In order to state our results we consider an action set $\mathcal{A}_{\mathrm{tree}}$ indexed by the nodes of a height $L$ binary tree defined here as having $L$ levels and $2^L - 1$ nodes. We call $a_{l,i}$ the $i$-th action of the $l$-th level of the tree. For an example see Figure 1. Let $\varepsilon > 0$. We define a function class $\mathcal{F}_{\mathrm{tree}}$ indexed by paths from the root node to a leaf. For any such path $\mathbf{p} = \{a_{1,1}, a_{2,i_2}, \cdots, a_{L,i_L}\}$ the function $f^{(\mathbf{p})}$ equals,

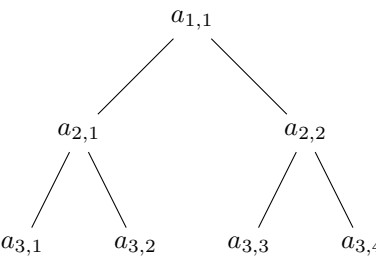

Figure 1: Binary Tree

$$f^{(\mathbf{p})}(a) = \begin{cases} 1 & \text{if } a = a_{L,i_L} \\ 1 - 2\varepsilon & \text{if } a \in \mathbf{p}\backslash\{a_{L,i_L}\} \\ 1 - 12\varepsilon & \text{o.w.} \end{cases}$$

The following result fleshes out a separation between planning and adaptive learning with action set $\mathcal{A}_{\mathrm{tree}}$ and function class $\mathcal{F}_{\mathrm{tree}}$ in the setting where at time $T$ the learner will produce a guess for the optimal policy $\hat{\pi}_T$.

**Theorem 6.1.** *Let $\varepsilon > 0$, $T \in \mathbb{N}$. Consider the action set $\mathcal{A}_{\mathrm{tree}}$ and function class $\mathcal{F}_{\mathrm{tree}}$ and a reward noise process such that $\xi_t \sim \mathcal{N}(0,1)$ conditionally for all $t \in [T]$. For any planning algorithm $\mathrm{Alg}$ there is a function $f_\star \in \mathcal{F}_{\mathrm{tree}}$ such that when $T \leqslant \frac{2^{L-5}}{9\varepsilon^2}$ and $\mathrm{Alg}$ interacts with $f_\star$ then,*

$$\mathbb{E}_{\mathrm{Alg}, f_\star}\left[\mathbb{E}_{a \sim \hat{\pi}_T}\left[f_\star(a)\right]\right] < \max_{a \in \mathcal{A}_{\mathrm{tree}}} f_\star(a) - \varepsilon.$$

*Moreover, there is an adaptive algorithm $\mathrm{Alg}_{\mathrm{adaptive}}$ such that if $T \geqslant \frac{2L \log(2L/\varepsilon)}{\varepsilon^2}$,*

$$\mathbb{E}_{\mathrm{Alg}_{\mathrm{adaptive}}}\left[\mathbb{E}_{a \sim \hat{\pi}_T}\left[f(a)\right]\right] \geqslant \max_{a \in \mathcal{A}_{\mathrm{tree}}} f(a) - \varepsilon.$$

*for all $f \in \mathcal{F}_{\mathrm{tree}}$. Where $\mathbb{E}_{\widetilde{\mathrm{Alg}}, \widetilde{f}}\left[\cdot\right]$ is the expectation over the randomness of $\widetilde{\mathrm{Alg}}$ and the environment for target function $\widetilde{f}$, and $\hat{\pi}_T$ is the algorithm's final policy guess after the sampling phase.*

---

[5]We obviate the $T$ dependence on $d_{\mathrm{eluder}}$ for readability.

The main insight behind Theorem 6.1 is that adaptive strategies to find an optimal action in the $\mathcal{F}_{\text{tree}}$ function class can make use of the conditional structure of the action space by trying to determine a path of actions from the root to the leaf containing the optimal action. An adaptive strategy can determine this path by querying only nodes that are adjacent to it. In contrast, a static experiment planning strategy cannot leverage this structure and has to query all leaf nodes. Theorem 6.1 implies a gap between the adaptive and experiment planning problems. Moreover, since the eluder dimension of $\mathcal{F}_{\text{tree}}$ scales with $2^L$ (see Lemma D.1), OptLS and SquareCB are suboptimal adaptive algorithms for this model class. In contrast with the $\mathcal{O}\left(\frac{2L\log(2L/\varepsilon)}{\varepsilon^2}\right)$ upper bound in Theorem 6.1, converting the cummulative regret bounds of OptLS and SquareCB yield guarantees scaling as $\mathcal{O}\left(\frac{2^L\log(|\mathcal{F}_{\text{tree}}|)}{\varepsilon^2}\right)$.

# 7 Model Selection

In this section we consider a setting where the learner has access to $\mathcal{F}_1, \cdots, \mathcal{F}_M$ function (model) classes all with domain $\mathcal{X} \times \mathcal{A}$. The learner is promised there exists an index $i_\star$ such that $f_\star \in \mathcal{F}_{i_\star}$. The value of index $i_\star$ is not known by the learner. We will show the uniform sampling strategy has a sample complexity that scales logarithmically with the number of models and with the complexity of the optimal model class $|\mathcal{F}_{i_\star}|$. In this setting the Sampler will collect two uniform actions datasets $(\widetilde{\mathcal{D}}_T^{(\text{Train})}, \widetilde{\mathcal{D}}_T^{(\text{Test})})$ of size $T/2$ each. Using the "train" dataset $\widetilde{\mathcal{D}}_T^{(\text{Train})}$ the learner computes candidate reward models $\widehat{f}_T^{(i)}$ for all $i \in [M]$ by solving the least squares problems:

$$\widehat{f}_T^{(i)} = \arg\min_{f \in \mathcal{F}_i} \sum_{(x_i, a_i, r_i) \in \widetilde{\mathcal{D}}_T^{(\text{Train})}} (f(x_i, a_i) - r_i)^2.$$

Using the "test" dataset the learner extracts a guess for the optimal model class index $\underline{i}$ by solving,

$$\underline{i} = \arg\min_{i \in [M]} \sum_{(x_\ell, a_\ell, r_\ell) \in \widetilde{\mathcal{D}}_T^{(\text{Test})}} \left(\widehat{f}_T^{(i)}(x_\ell, a_\ell) - r_\ell\right)^2. \tag{7}$$

The candidate policy $\widehat{\pi}_T$ is defined as $\widehat{\pi}_T(x) = \arg\max_{a \in \mathcal{A}} \widehat{f}_T^{(\underline{i})}(x, a)$. Let's start by relating the expected least squares loss of the candidate model $\widehat{f}_T^{(\underline{i})}$ to the size of the optimal model class $|\mathcal{F}_{i_\star}|$,

**Proposition 7.1.** *There exists a universal constant $\underline{c} > 0$ such that,*

$$\mathbb{E}_{x \sim \mathcal{P}, a \sim \text{Uniform}(\mathcal{A})}\left[\left(f_\star(x, a) - \widehat{f}_T^{(\underline{i})}(x, a)\right)^2\right] \leq \frac{\underline{c}\max^2(B, \bar{B})\log(T\max(M, |\mathcal{F}_{i_\star}|)/\delta)}{T}.$$

*With probability at lest $1 - \delta$.*

The proof of Proposition 7.1 can be found in Appendix E.1. The uniform sampling model-selection algorithm has the following performance guarantees,

**Theorem 7.2.** *There exists a universal constant $\widetilde{c}' > 0$ s.t. if $T \geq \widetilde{c}' \frac{\max^2(B, \bar{B})|\mathcal{A}|\log\left(\frac{\max(M, |\mathcal{F}_\star|)}{\varepsilon\delta}\right)}{\varepsilon^2}$ then with probability at least $1 - \delta$, the candidate policy $\widehat{\pi}_T$ of the uniform sampling model-selection algorithm is $\varepsilon-$optimal.*

*Proof.* Due to the realizability assumption ($r_t = f_\star(x_t, a_t) + \xi_t$) the instantaneous regret of $\widehat{\pi}_T$ on context $x \in \mathcal{X}$ equals $\max_a f_\star(x, a) - f_\star(x, \widehat{\pi}_T(x))$. Let $\pi_\star(x) = \arg\max_{a \in \mathcal{A}} f_\star(x, a)$. Just like in the proof of Theorem 5.1 (see Equation 25), we can relate the suboptimality of $\widehat{\pi}_T$ with the expected least squares loss of $\widehat{f}_T^{(\underline{i})}$ under the uniform policy,

$$\mathbb{E}_{x \sim \mathcal{P}}\left[\max_{a \in \mathcal{A}} f_\star(x, a) - f_\star(x, \widehat{\pi}_T(x))\right] \leq 2\sqrt{|\mathcal{A}|\mathbb{E}_{x \sim \mathcal{P}, a \sim \text{Uniform}(\mathcal{A})}\left[\left(f_\star(x, a) - \widehat{f}_T^{(\underline{i})}(x, a)\right)^2\right]}. \tag{8}$$

Finally Proposition 7.1 implies thre is a universal constant $\underline{c} > 0$ such that,

$$\mathbb{E}_{x \sim \mathcal{P}, a \sim \text{Uniform}(\mathcal{A})}\left[\left(f_\star(x, a) - \widehat{f}_T^{(\underline{i})}(x, a)\right)^2\right] \leq \frac{\underline{c}(\bar{B}^2 + B^2)\log(T\max(M, |\mathcal{F}_{i_\star}|)/\delta)}{T}$$

with probability at least $1 - \delta$. Plugging this result back into Equation 8 we see the suboptimality of $\widehat{\pi}_T$ can be upper bounded as,

$$\mathbb{E}_{x \sim \mathcal{P}} \left[ \max_{a \in \mathcal{A}} f_\star(x, a) - f_\star(x, \widehat{\pi}_T(x)) \right] \leqslant 2\sqrt{\frac{\underline{c}|\mathcal{A}|(\bar{B}^2 + B^2) \log(T \max(M, |\mathcal{F}_{i_\star}|)/\delta)}{T}}$$

Setting $g(T) = 4\frac{\underline{c}|\mathcal{A}|(\bar{B}^2 + B^2) \log(T \max(M, |\mathcal{F}_{i_\star}|)/\delta)}{T}$ in Lemma A.6 implies there exists a universal

constant $\tilde{c}' > 0$ such that $g(T) \leqslant \varepsilon^2$ for all $T \geqslant \tilde{c}' \frac{\max^2(B,\bar{B})|\mathcal{A}| \log\left(\frac{\max(M, |\mathcal{F}_{i_\star}|)}{\varepsilon\delta}\right)}{\varepsilon^2}$. $\qquad \square$

The results of Theorem 7.2 can be best understood by contrasting them to the uniform sampling algorithm with input model class equal to the union of the function classes $\mathcal{F}_{\mathrm{all}} = \cup_{i \in [M]} \mathcal{F}_i$. Applying the results of Theorem 5.1 to $\mathcal{F}_{\mathrm{all}}$ yields a sample complexity scaling with $\max_{i \in [M]} \log(|\mathcal{F}|_i)$, a quantity that could be much larger than $\log(|\mathcal{F}_{i_\star}|)$. In contrast, the uniform sampling model-selection algorithm achieves a sample complexity scaling with $\log(|\mathcal{F}_{i_\star}|)$ at a price logarithmic in the number of models classes $M$. This logarithmic dependence on $M$ stands apart from model selection algorithms for cumulative regret scenarios such as Corral [5, 39], ECE [30] and RegretBalancing [38, 12] that instead have a polynomial dependence on $M$. The uniform sampling model-selection algorithm is agnostic to the value of $\varepsilon$. The results of Theorem 7.2 hold for any $\varepsilon$. If $\varepsilon$ is known in advance the learner can compute the model class index $\widehat{i} = \arg\min \left\{ i \in [M] \text{ s.t. } T \geqslant \tilde{c} \frac{\max^2(B,\bar{B})|\mathcal{A}| \log(|\mathcal{F}_i|/\varepsilon\delta)}{\varepsilon^2} \right\}$ and use the uniform sampling strategy for $\mathcal{F}_{\widehat{i}}$. For this choice of $\varepsilon$, Theorem 5.1 guarantees similar bounds to those of Theorem 7.2. Unfortunately in contrast with the uniform sampling model-selection algorithm this method would be valid for a single choice of $\varepsilon$.

# 8 Conclusion

In this work we have introduced the first set of algorithms for the experiment planning problem for contextual bandits with general function approximation. We have developed the EluderPlanning algorithm that produces a static policy sequence that after deployment can be used to recover an $\varepsilon-$optimal policy. We showed it is enough for the number of static policies and therefore samples during the sampling phase to be as large as the number of samples required from an adaptive procedure based on an online-to-batch conversion of the OptLS algorithm. Similarly we also demonstrated the uniform sampling strategy enjoys the same online-to-batch conversion sample complexity as the SquareCB algorithm. These results seemingly suggest that simple regret rates for adaptive learning may also be achieved in experiment planning scenarios. We show this is not the case. There exist structured bandit problems for which adaptive learning may require a number of samples that is substantially smaller than the number of samples required by a static policy sequence. This is significant because it implies the suboptimality of the rates achieved by existing adaptive learning algorithms such as OptLS and SquareCB and also because it draws a clear distinction between adaptive learning and experiment planning. This implies these algorithms are either suboptimal or their upper bound analysis is not tight. We believe the first to be correct. This is an important open question we hope to see addressed in future research. We have also introduced the first model selection results for the experiment planning problem.

Many important questions remain regarding this setting. Chief among them is to characterize the true statistical complexity of experimental design for contextual bandits with general function approximations. Our results indicate the eluder dimension is not the sharpest statistical complexity measure to characterize learning here. Developing a more new form of complexity, as well as an accompanying algorithm that can achieve the true statistical lower bound for the problem of experiment planning remains an exciting and important open question to tackle in future research.

## Acknowledgments and Disclosure of Funding

We thank Akshay Krishnamurthy for helpful discussions. Aldo Pacchiano would like to thank the support of the Eric and Wendy Schmidt Center at the Broad Institute of MIT and Harvard. This work was supported in part by funding from the Eric and Wendy Schmidt Center at the Broad Institute of MIT and Harvard. This work was supported in part by NSF grant #2112926.

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
