# A    Supporting Lemmas

**Lemma A.1** (Hoeffding Inequality). *Let $\{Y_\ell\}_{\ell=1}^\infty$ be a martingale difference sequence such that $Y_\ell$ is $Y_\ell \in [a_\ell, b_\ell]$ almost surely for some constants $a_\ell, b_\ell$ almost surely for all $\ell = 1, \cdots, t$. then*

$$\sum_{\ell=1}^t Y_\ell \leqslant 4 \sqrt{\sum_{\ell=1}^t (b_\ell - a_\ell)^2 \log\left(\frac{1}{\widetilde{\delta}}\right)}$$

*with probability at least $1 - \widetilde{\delta}$.*

**Lemma A.2** (Anytime Hoeffding Inequality [38]). *Let $\{Y_\ell\}_{\ell=1}^\infty$ be a martingale difference sequence such that $Y_\ell$ is $Y_\ell \in [a_\ell, b_\ell]$ almost surely for some constants $a_\ell, b_\ell$ almost surely for all $\ell = 1, \cdots, t$. then*

$$\sum_{\ell=1}^t Y_\ell \leqslant 2 \sqrt{\sum_{\ell=1}^t (b_\ell - a_\ell)^2 \log\left(\frac{12t^2}{\widetilde{\delta}}\right)}$$

*with probability at least $1 - \widetilde{\delta}$ for all $t \in \mathbb{N}$ simultaneously.*

Our results relies on the following variant of Bernstein inequality for martingales, or Freedman's inequality [19], as stated in e.g., [3, 7].

**Lemma A.3** (Simplified Freedman's inequality). *Let $X_1, ..., X_T$ be a bounded martingale difference sequence with $|X_\ell| \leqslant R$. For any $\delta' \in (0, 1)$, and $\eta \in (0, 1/R)$, with probability at least $1 - \delta'$,*

$$\sum_{\ell=1}^T X_\ell \leqslant \eta \sum_{\ell=1}^T \mathbb{E}_\ell[X_\ell^2] + \frac{\log(1/\delta')}{\eta}. \tag{9}$$

*where $\mathbb{E}_\ell[\cdot]$ is the conditional expectation[6] induced by conditioning on $X_1, \cdots, X_{\ell-1}$.*

**Lemma A.4** (Anytime Freedman). *Let $\{X_t\}_{t=1}^\infty$ be a bounded martingale difference sequence with $|X_t| \leqslant R$ for all $t \in \mathbb{N}$. For any $\delta' \in (0, 1)$, and $\eta \in (0, 1/R)$, there exists a universal constant $C > 0$ such that for all $t \in \mathbb{N}$ simultaneously with probability at least $1 - \delta'$,*

$$\sum_{\ell=1}^t X_\ell \leqslant \eta \sum_{\ell=1}^t \mathbb{E}_\ell[X_\ell^2] + \frac{C \log(t/\delta')}{\eta}. \tag{10}$$

*where $\mathbb{E}_\ell[\cdot]$ is the conditional expectation induced by conditioning on $X_1, \cdots, X_{\ell-1}$.*

*Proof.* This result follows from Lemma A.3. Fix a time-index $t$ and define $\delta_t = \frac{\delta'}{12t^2}$. Lemma A.3 implies that with probability at least $1 - \delta_t$,

$$\sum_{\ell=1}^t X_\ell \leqslant \eta \sum_{\ell=1}^t \mathbb{E}_\ell\left[X_\ell^2\right] + \frac{\log(1/\delta_t)}{\eta}.$$

A union bound implies that with probability at least $1 - \sum_{\ell=1}^t \delta_t \geqslant 1 - \delta'$,

$$\sum_{\ell=1}^t X_\ell \leqslant \eta \sum_{\ell=1}^t \mathbb{E}_\ell\left[X_\ell^2\right] + \frac{\log(12t^2/\delta')}{\eta}$$

$$\overset{(i)}{\leqslant} \eta \sum_{\ell=1}^t \mathbb{E}_\ell\left[X_\ell^2\right] + \frac{C \log(t/\delta')}{\eta}.$$

holds for all $t \in \mathbb{N}$. Inequality $(i)$ holds because $\log(12t^2/\delta') = \mathcal{O}\left(\log(t\delta')\right)$.

$\square$

---

[6]We will use this notation to denote conditional expectations throughout this work.

**Lemma A.5.** *Let $\{A_\ell\}_{\ell=1}^\infty$ be an adapted process. We use the notation $\mathbb{E}_\ell[\cdot]$ to denote the conditional expectation $\mathbb{E}[\cdot|A_1, \cdots, A_{\ell-1}]$. If $|A_\ell| \leqslant \tilde{B}$ for all $\ell \in \mathbb{N}$ a.s. then*

$$\sum_{\ell=1}^t A_\ell^2 \leqslant \frac{3}{2} \sum_{\ell=1}^t \mathbb{E}_\ell[A_\ell^2] + 2C\tilde{B}^2 \log\left(\frac{t}{\delta}\right).$$

*and*

$$\frac{\sum_{\ell=1}^t \mathbb{E}_\ell[A_\ell^2]}{2} \leqslant \sum_{\ell=1}^t A_\ell^2 + 2C\tilde{B}^2 \log\left(\frac{t}{\delta}\right)$$

*With probability at least $1 - \delta$, for all $t \in \mathbb{N}$ simultaneously where $C > 0$ is a universal constant.*

*Proof.* Consider a martingale difference sequence $\{W_\ell\}_{\ell=1}^\infty$ defined as $W_\ell = A_\ell^2 - \mathbb{E}_\ell[A_\ell^2]$. By definition,

$$|W_\ell| \leqslant \tilde{B}^2, \text{ and } \mathbb{E}_\ell[W_\ell^2] \leqslant \mathbb{E}_\ell[A_\ell^4] \leqslant \tilde{B}^2 \mathbb{E}_\ell[A_\ell^2].$$

for all $\ell \in \mathbb{N}$.

The anytime freedman inequality (Lemma A.4) and a union bound on the martingale sequences $\{W_\ell\}_{\ell=1}^\infty$ and $\{-W_\ell\}_{\ell=1}^\infty$ implies

$$\left|\sum_{\ell=1}^t W_\ell\right| \leqslant \eta \sum_{\ell=1}^t \mathbb{E}_\ell[W_\ell^2] + \frac{C\log(2t/\delta)}{\eta}$$

$$\overset{(i)}{\leqslant} \eta\tilde{B}^2 \sum_{\ell=1}^t \mathbb{E}_\ell[A_\ell^2] + \frac{C\log(2t/\delta)}{\eta}$$

for all $t \in \mathbb{N}$ with probability at least $1 - \delta$. Inequality $(i)$ follows because $\mathbb{E}_\ell[W_\ell^2] \leqslant \mathbb{E}_\ell[A_\ell^4] \leqslant \tilde{B}^2 \mathbb{E}_\ell[A_\ell^2]$. Setting $\eta = \frac{1}{2\tilde{B}^2} < \frac{1}{\tilde{B}^2}$.

$$\left|\sum_{\ell=1}^t W_\ell\right| \leqslant \frac{\sum_{\ell=1}^t \mathbb{E}_\ell[A_\ell^2]}{2} + 2C\tilde{B}^2 \log(2t/\delta) \overset{(i)}{\leqslant} \frac{\sum_{\ell=1}^t \mathbb{E}_\ell[A_\ell^2]}{2} + C'\tilde{B}^2 \log(t/\delta)$$

where inequality $(i)$ follows from $\log(2t/\delta) = \mathcal{O}(\log(t/\delta))$. Substituting $\left|\sum_{\ell=1}^t W_\ell\right|$ with $-\sum_{\ell=1}^t W_\ell$ and $\sum_{\ell=1}^t W_\ell$ and simplifying the resulting expressions yields the result. $\square$

**Lemma A.6.** *Let $c > 0, \alpha \geqslant 1, \xi \in (0, 1]$ and $g : \mathbb{R} \to \mathbb{R}$ be defined as $g(T) = c\frac{\log^\alpha(T)}{T}$. Let $\xi' = \frac{\xi}{c(\alpha+4)^\alpha}$. For all $T \geqslant \frac{\log^\alpha(1/\xi')}{\xi'}$*

$$g(T) \leqslant \xi.$$

*Proof.* Let's see that $g(T)$ is an increasing function for all $T \geqslant$. The derivative of $g(T)$ satisfies $\frac{\partial g(T)}{\partial T} = \frac{\log^{\alpha-1}(\alpha - \log(T))}{T^2}$. Hence for all $T$ such that $\log(T) \geqslant \alpha$ (i.e. $T \geqslant e^\alpha$), the derivative is negative and therefore $f$ is decreasing.

Let $T_0 = \frac{\log^\alpha(1/\xi')}{\xi'}$. Substituting $g(T_0)$ we get,

$$g(T_0) = \frac{c\log^\alpha\left(\frac{\log^\alpha(1/\xi')}{\xi'}\right)}{\log^\alpha(1/\xi')}\xi'$$

The numerator satisfies,

$$
\begin{aligned}
\log^{\alpha}\left(\frac{\log^{\alpha}(1/\xi')}{\xi'}\right) &= \left(\log\left(\log^{\alpha}(1/\xi')\right) + \log(1/\xi')\right)^{\alpha} \\
&= \left(\alpha\log(\log(1/\xi')) + \log(1/\xi')\right)^{\alpha} \\
&\overset{(i)}{\leqslant} \left((\alpha+1)\log(1/\xi')\right)^{\alpha} \\
&= (\alpha+1)^{\alpha}\log^{\alpha}(1/\xi')
\end{aligned}
$$

Inequality $(i)$ holds because $\log(1/\xi') \leqslant 1/\xi'$ since $\xi' \in (0,1]$. Substituting these inequalities in the expression above yields,

$$
g(T_0) \leqslant c(\alpha+1)^{\alpha}\xi'.
$$

Setting $\xi' = \frac{\xi}{c(\alpha+4)^{\alpha}}$ and noting this definition of $T_0$ satisfies $T_0 \geqslant e^{\alpha}$ yields the desired result.

$\square$

## A.1 Conditions for $\beta_{\mathcal{F}}(t,\delta)$

**Proposition A.7.** *There exists a universal constant $\bar{C}$ such that $\beta_{\mathcal{F}}(\delta,t) = \bar{C}(B+\bar{B})\sqrt{\log\left(\frac{|\mathcal{F}|t}{\delta}\right)}$ satisfies*

$$
\beta_{\mathcal{F}}^2(\delta,t) \geqslant 4(2C_1^2+C_2)B^2\log\left(\frac{2t|\mathcal{F}|}{\delta}\right) \quad \text{and} \quad \beta_{\mathcal{F}}^2(\delta,t) \geqslant \Omega\left((B^2+\bar{B}^2)\log\left(\frac{|\mathcal{F}|t}{\delta}\right)\right)
$$

*for all $t \in \mathbb{N}$ and $\delta \in (0,1)$. Where $C_1, C_2$ are the universal constants from equation 23 in Lemma 4.6 (left) and $\beta_{\mathcal{F}}^2(\delta,t) \geqslant C_{t,\delta}$ in Lemma A.8 (right).*

*Proof.* This result follows immediately from the definition. $\square$

Satisfies both conditions.

## A.2 Least Squares Guarantees

In this section we provide bounds for the least squares algorithm. We assume a setting where at each step $t$ the learner observes an element $z_t \in \mathcal{Z}$ with a label $y_t = f_\star(z) + \xi_t$ where $f_\star \in \mathcal{F}$, $\max_{f \in \mathcal{F}, z \in \mathcal{Z}}|f(z)| \leqslant B$ and $\xi_i t$ is a conditionally zero mean $1-$subgaussian random variable satisfying $|\xi_t| \leqslant \bar{B}$. The function $f_\star$ is assumed to satisfy $f_\star \in \mathcal{F}$ for a known function class $\mathcal{F}$. We analyze the least squares procedure,

$$
\widehat{f}_t = \min_{f \in \mathcal{F}} \sum_{\ell=1}^{t-1} \left(f(z_\ell) - y_\ell\right)^2.
$$

**Lemma A.8** (LS guarantee). *Let $z_1, \cdots, z_T$ be a sequence of queries and values $y_1, \cdots, y_T$. Define $\widehat{f}_t = \arg\min_{f \in \mathcal{F}} \sum_{\ell=1}^{t-1}\left(f(z_\ell) - y_\ell\right)^2$ for a function class $\mathcal{F}$ satisfying $\max_{f \in \mathcal{F}, z \in \mathcal{Z}}|f(z)| \leqslant B$ and where $y_\ell(z_\ell) = f_\star(z_\ell) + \xi_\ell$ for $\xi_\ell$ a conditionally zero mean random variable with $|\xi_i| \leqslant \bar{B}$ then,*

$$
\sum_{\ell=1}^{t-1}\left(\widehat{f}_t(z_\ell) - f_\star(z_\ell)\right)^2 \leqslant C_{t,\delta} \leqslant C_{T,\delta}.
$$

*Moreover, if $z_t = (x_t, a_t)$ with $x_t \sim \mathcal{P}_t$ and $a_t \sim \pi_t(\cdot|x_t)$ an adapted probability-policy sequence,*

$$
\sum_{\ell=1}^{t-1}\mathbb{E}_{x \sim \mathcal{P}_\ell, a \sim \pi_\ell(\cdot|x)}\left[\left(\widehat{f}_t(z_\ell) - f_\star(z_i)\right)^2\right] \leqslant C_{t,\delta} \leqslant C_{T,\delta}.
$$

*with probability at least $1-\delta$ for all $t \in \mathbb{N}$ where $C_{t,\delta} = \mathcal{O}\left((B^2+\bar{B}^2)\log\left(\frac{|\mathcal{F}|t}{\delta}\right)\right)$.*

*Proof.* By definition $\mathbb{E}_\ell[\xi_\ell] = 0$ and $\mathbb{E}_\ell[\xi_\ell^2] \leq \bar{B}^2$ where we define these conditional expectations to include all events occuring before $\xi_\ell$ (including $z_\ell$). Since $\widehat{f}_t$ is the empirical minimizer of the least squares loss,

$$\sum_{\ell=1}^{t-1} \left( \widehat{f}_t(z_\ell) - f_\star(z_\ell) - \xi_\ell \right)^2 \leq \sum_{\ell=1}^{t} \xi_\ell^2.$$

And therefore,

$$\sum_{\ell=1}^{t-1} \left( \widehat{f}_t(z_\ell) - f_\star(z_\ell) \right)^2 \leq 2 \sum_{\ell=1}^{t-1} \xi_\ell \cdot \left( \widehat{f}_t(z_\ell) - f_\star(z_\ell) \right). \tag{11}$$

For any fixed $f$ let's consider the martingale difference sequence $Z_\ell^f = \xi_\ell (f(z_\ell) - f_\star(z_\ell))$. Observe that $|Z_\ell^f| \leq 2\bar{B}B$ and that,

$$\mathbb{E}_\ell \left[ \left( Z_\ell^f \right)^2 \right] \leq \bar{B}^2 \left( f(z_\ell) - f_\star(z_\ell) \right)^2. \tag{12}$$

Recall that $\mathbb{E}_\ell[\cdot]$ conditions on all events right before $\xi_\ell$ including $z_\ell$ so that $\mathbb{E}_\ell[(f(z_\ell) - f_\star(z_\ell)^2] = (f(z_\ell) - f_\star(z_\ell)^2$. By the Anytime Friedman's inequality (see Lemma A.4) with $\eta = \frac{1}{4B\max(B,\bar{B})} < \min\left( \frac{1}{2\bar{B}B}, \frac{1}{2\bar{B}^2} \right)$ and $\delta' = \frac{\delta}{2|\mathcal{F}|}$, and a union bound over all $f \in \mathcal{F}$

$$\sum_{\ell=1}^{t-1} \xi_\ell \cdot (f(z_\ell) - f_\star(z_\ell)) \leq \eta \sum_{\ell=1}^{t-1} \mathbb{E}_\ell \left[ \left( Z_\ell^f \right)^2 \right] + \frac{C\log(2|\mathcal{F}|t/\delta)}{\eta}$$

$$\overset{(i)}{\leq} \eta \bar{B}^2 \sum_{\ell=1}^{t-1} (f(z_\ell) - f_\star(z_\ell))^2 + \frac{C\log(2|\mathcal{F}|t/\delta)}{\eta}$$

$$\overset{(ii)}{\leq} \frac{\sum_{\ell=1}^{t-1} (f(z_\ell) - f_\star(z_\ell))^2}{4} + \mathcal{O}\left( (\bar{B}^2 + \bar{B}B) \log\left( \frac{|\mathcal{F}|t}{\delta} \right) \right)$$

for all $f \in \mathcal{F}$ and all $t \in \mathbb{N}$ simultaneously with probability at least $1 - \delta/2$. Where inequality $(i)$ holds because of Equation 12 and inequality $(ii)$ holds because $\eta\bar{B}^2 \leq \frac{1}{4}$. In particular,

$$\sum_{\ell=1}^{t-1} \xi_\ell \cdot \left( \widehat{f}_t(z_\ell) - f_\star(z_\ell) \right) \leq \frac{\sum_{\ell=1}^{t-1} \left( \widehat{f}_t(z_\ell) - f_\star(z_\ell) \right)^2}{4} + \mathcal{O}\left( (\bar{B}^2 + \bar{B}B) \log\left( \frac{|\mathcal{F}|t}{\delta} \right) \right) \tag{13}$$

Combining inequalities 11 and 13 we conclude,

$$\sum_{\ell=1}^{t-1} \left( \widehat{f}_t(z_\ell) - f_\star(z_\ell) \right)^2 \leq \mathcal{O}\left( \bar{B}\max(B, \bar{B}) \log\left( \frac{|\mathcal{F}|t}{\delta} \right) \right). \tag{14}$$

for all $f \in \mathcal{F}$ and all $t \in \mathbb{N}$ simultaneously with probability at least $1 - \delta/2$. This finalizes the first part of the result.

When $z_t = (x_t, a_t)$ with $x_t \sim \mathcal{P}_t$ and $a_t \sim \pi_t(\cdot|x_t)$ Lemma A.5 applied to the adapted sequence $\{A_\ell\}_{\ell=1}^\infty$ where $A_\ell = f(z_\ell) - f_\star(z_\ell)$ implies,

$$\frac{1}{2} \sum_{\ell=1}^{t-1} \mathbb{E}_{x \sim \mathcal{P}_\ell, a \sim \pi_\ell(\cdot|x)}[(f(z_\ell) - f_\star(z_\ell))^2] \leq \sum_{\ell=1}^{t-1} (f(z_\ell) - f_\star(z_\ell))^2 + \mathcal{O}\left( B^2 \log\left( \frac{|\mathcal{F}|t}{\delta} \right) \right)$$

for all $f \in \mathcal{F}$ and all $t \in \mathbb{N}$ simultaneously with probability at least $1 - \delta/2$. In particular setting $f = \widehat{f}_t$ in the equation above,

$$\frac{1}{2} \sum_{\ell=1}^{t-1} \mathbb{E}_{x \sim \mathcal{P}_\ell, a \sim \pi_\ell(\cdot|x)}[\left( \widehat{f}_t(z_\ell) - f_\star(z_\ell) \right)^2] \leq \sum_{\ell=1}^{t-1} \left( \widehat{f}_t(z_\ell) - f_\star(z_\ell) \right)^2 + \mathcal{O}\left( B^2 \log\left( \frac{|\mathcal{F}|t}{\delta} \right) \right)$$

$$\tag{15}$$

Combining inequalities 14 and 15 we conclude,

$$\sum_{\ell=1}^{t-1} \mathbb{E}_{x \sim \mathcal{P}_t, a \sim \pi_t(\cdot|x)}[(f(z_\ell) - f_\star(z_\ell))^2] \leqslant \mathcal{O}\left((B^2 + \bar{B}^2)\log\left(\frac{|\mathcal{F}|t}{\delta}\right)\right).$$

for all $t \in \mathbb{N}$ simultaneously. This finalizes the second part of the result. A union bound finalizes the Lemma's proof. $\qquad \square$

**Lemma A.9.** *Assume a learner interacts with a function class $\mathcal{F}$ while satisfying realizability (Assumption 3.1), Bounded Function Range (Assumption 3.2) and Bounded Noise (Assumption 3.3). Let $f \in \mathcal{F}$ be an arbitrary (fixed) function and $\widetilde{\mathcal{D}}_T^{(\mathrm{Test})}$ be a size $T$ dataset produced by sampling $T$ i.i.d. contexts from $\mathcal{P}$ and acting according to the uniform policy* Uniform. *There is a universal constant $\underline{C} > 0$ such that,*

$$\mathbb{E}_{x \sim \mathcal{P}, a \sim \mathrm{Uniform}(\mathcal{A})}\left[(f_\star(x,a) - f(x,a))^2\right]$$

$$\leqslant \frac{4}{T}\left(\sum_{(x_\ell, a_\ell, r_\ell) \in \widetilde{\mathcal{D}}_T^{(\mathrm{Test})}} (f(x_\ell, a_\ell) - r_\ell)^2 - \xi_\ell^2\right) + \frac{\underline{C}(\bar{B}^2 + B^2)\log(T/\delta)}{T}$$

$$\leqslant 16\mathbb{E}_{x \sim \mathcal{P}, a \sim \mathrm{Uniform}(\mathcal{A})}\left[(f_\star(x,a) - f(x,a))^2\right] + 4\frac{\underline{C}(\bar{B}^2 + B^2)\log(T/\delta)}{T}.$$

*with probability at least $1 - \delta$.*

*Proof.* Throughout this proof we refer to the elements of $\widetilde{\mathcal{D}}_T^{(\mathrm{Test})}$ as the ordered list $x_1, a_1, \cdots, x_T, a_T$. Let's write $r_\ell = f_\star(x_\ell, a_\ell) + \xi_\ell$. Substituting this into the empirical loss we obtain,

$$\sum_{\ell=1}^{T} (f(x_\ell, a_\ell) - r_\ell)^2 = \sum_{\ell=1}^{T} (f(x_\ell, a_\ell) - f_\star(x_\ell, a_\ell))^2 + 2\xi_\ell(f(x_\ell, a_\ell) - f_\star(x_\ell, a_\ell)) + \xi_\ell^2.$$

Therefore,

$$\frac{1}{T}\left(\sum_{\ell=1}^{T}(f(x_\ell, a_\ell) - f_\star(x_\ell, a_\ell))^2 + 2\xi_\ell(f(x_\ell, a_\ell) - f_\star(x_\ell, a_\ell))\right)$$

$$= \frac{1}{T}\sum_{(x_\ell, a_\ell, r_\ell) \in \widetilde{\mathcal{D}}_T^{(\mathrm{Test})}} (f(x_\ell, a_\ell) - r_\ell)^2 - \xi_\ell^2. \quad (16)$$

Consider the martingale difference sequence $\{Z_\ell\}_{\ell=1}^{T}$ defined as $Z_\ell = \xi_\ell(f_\star(x_\ell, a_\ell) - f(x_\ell, a_\ell))$. Notice that $|Z_\ell| \leqslant 2\bar{B}B$ and that $\mathbb{E}_\ell[(\xi_\ell(f(x_\ell, a_\ell) - f_\star(x_\ell, a_\ell)))^2] \leqslant \bar{B}^2(f(x_\ell, a_\ell) - f_\star(x_\ell, a_\ell))^2$. The Anytime Freedman inequality (Lemma A.4) with $\eta = \frac{1}{4\bar{B}\max(\bar{B}, B)} < \min\left(\frac{1}{2\bar{B}B}, \frac{1}{2\bar{B}^2}\right)$ and $\delta' = \delta/2$ plus a union bound implies that with probability at least $1 - \delta/2$

$$\max\left(\sum_{\ell=1} Z_\ell, -\sum_{\ell=1} Z_\ell\right) \leqslant \eta\sum_{\ell=1}^{T} \mathbb{E}_\ell[(\xi_\ell(f(x_\ell, a_\ell) - f_\star(x_\ell, a_\ell))^2] + \frac{C\log(2T/\delta)}{\eta}$$

$$\leqslant \eta\bar{B}^2\sum_{\ell=1}^{T}(f(x_\ell, a_\ell) - f_\star(x_\ell, a_\ell))^2 + \frac{C\log(2T/\delta)}{\eta} \quad (17)$$

$$\overset{(i)}{\leqslant} \frac{1}{4}\sum_{\ell=1}^{T}(f(x_\ell, a_\ell) - f_\star(x_\ell, a_\ell))^2 + C_3(\bar{B}^2 + B^2)\log(T/\delta). \quad (18)$$

where inequality $(i)$ follows because $\eta\bar{B}^2 2 \leqslant \frac{1}{4}$ and because $C\log(2T/\delta) = \mathcal{O}(\log(T/\delta))$. Substituting $Z_\ell = \xi_\ell(f_\star(x_\ell, a_\ell) - f(x_\ell, a_\ell))$, considering the inequality with $-\sum_{\ell=1} Z_\ell$ on the LHS of the above display and dividing by $T$ the expression above yields,

$$-\frac{1}{2T}\sum_{\ell=1}^{T}(f(x_\ell, a_\ell) - f_\star(x_\ell, a_\ell))^2 - \frac{2C_3}{T}(\bar{B}^2 + B^2)\log(T/\delta) \leqslant \frac{1}{T}\sum_{\ell=1}^{T} 2\xi_\ell(f(x_\ell, a_\ell) - f_\star(x_\ell, a_\ell))$$

and therefore, combining this expression above with Equation 16 yields,

$$\frac{1}{2T}\sum_{\ell=1}^{T}(f(x_\ell,a_\ell)-f_\star(x_\ell,a_\ell))^2 - \frac{2C_3}{T}(\bar{B}^2+B^2)\log(T/\delta)$$

$$\leq \frac{1}{T}\sum_{(x_\ell,a_\ell,r_\ell)\in\widetilde{\mathcal{D}}_T^{(\text{Test})}}(f(x_\ell,a_\ell)-r_\ell)^2 - \xi_\ell^2. \qquad (19)$$

Finally, the second item of Lemma A.5 applied to the $\{A_\ell\}_{\ell=1}^T$ sequence with $A_\ell = f_\star(x_\ell,a_\ell) - f_\star(x_\ell,a_\ell)$ implies (after dividing by $T$) and noting that at all time-steps the context and sampling distribution are the same,

$$\mathbb{E}_{x\sim\mathcal{P},a\sim\text{Uniform}(\mathcal{A})}\left[(f_\star(x,a)-f(x,a))^2\right]$$

$$\leq \frac{2}{T}\sum_{\ell=1}^{T}(f(x_\ell,a_\ell)-f_\star(x_\ell,a_\ell))^2 + \frac{C_4}{T}(\bar{B}^2+B^2)\log(T/\delta) \qquad (20)$$

and

$$\frac{1}{T}\sum_{\ell=1}^{T}(f(x_\ell,a_\ell)-f_\star(x_\ell,a_\ell))^2$$

$$\leq \frac{3}{2}\mathbb{E}_{x\sim\mathcal{P},a\sim\text{Uniform}(\mathcal{A})}\left[(f_\star(x,a)-f(x,a))^2\right] + \frac{C_4}{T}(\bar{B}^2+B^2)\log(T/\delta) \qquad (21)$$

with probability at least $1-\delta/2$. Combining Equations 19 and 20,

$$\mathbb{E}_{x\sim\mathcal{P},a\sim\text{Uniform}(\mathcal{A})}\left[(f_\star(x,a)-f(x,a))^2\right]$$

$$\leq \frac{4}{T}\sum_{(x_\ell,a_\ell,r_\ell)\in\widetilde{\mathcal{D}}_T^{(\text{Test})}}(f(x_\ell,a_\ell)-r_\ell)^2 - \xi_\ell^2 + \frac{\bar{\underline{C}}'(\bar{B}^2+B^2)\log(T/\delta)}{T}$$

with probability at least $1-\delta$ for some universal constant $\bar{\underline{C}}' > 0$.

To prove the second part of the argument we notice the following sequence of equalities and inequalities hold,

$$\frac{1}{T}\sum_{(x_\ell,a_\ell,r_\ell)\in\widetilde{\mathcal{D}}_T^{(\text{Test})}}(f(x_\ell,a_\ell)-r_\ell)^2 - \xi_\ell^2$$

$$= \frac{1}{T}\left(\sum_{\ell=1}^{T}(f(x_\ell,a_\ell)-f_\star(x_\ell,a_\ell))^2 + 2\xi_\ell(f(x_\ell,a_\ell)-f_\star(x_\ell,a_\ell))\right)$$

$$\overset{(i)}{\leq} \frac{1}{T}\left(\frac{3}{2}\sum_{\ell=1}^{T}(f(x_\ell,a_\ell)-f_\star(x_\ell,a_\ell))^2 + 2C_3(\bar{B}^2+B^2)\log(2T/\delta)\right)$$

$$\overset{(ii)}{\leq} \frac{9}{4}\mathbb{E}_{x\sim\mathcal{P},a\sim\text{Uniform}(\mathcal{A})}\left[(f_\star(x,a)-f(x,a))^2\right] + \frac{\bar{\underline{C}}''(\bar{B}^2+B^2)\log(T/\delta)}{T}$$

for some universal constant $\bar{\underline{C}}'' > 0$ and where inequality $(i)$ follows from equation 18 and considering the inequality with $-\sum_{\ell=1}^{T}Z_\ell$ on the RHS. Inequality $(ii)$ is a result of Equation 21.

Picking a universal constant $\bar{\underline{C}}$ larger than all the universal constants in this discussion plus a union bound finalizes the result.

$$\square$$

## B  Online to Batch Conversion

**Lemma B.1.** *Let $T\in\mathbb{N}$ and $\delta'\in[0,1]$. Assume there is an algorithm* Alg *that interacts sequentially with i.i.d. contexts $x_1,\cdots,x_T \overset{i.i.d.}{\sim} \mathcal{P}$ producing policies $\{\pi_\ell\}_{\ell=1}^T$, playing actions $a_\ell \sim \pi_\ell(x_\ell)$*

and observing rewards $r_\ell = f_\star(x_\ell a_\ell) + \xi_\ell$ for all $\ell \leqslant T$ and with $\xi_\ell$ a conditionally zero mean $1-$subgaussian random variable such that with probability at least $1 - \delta'$,

$$\sum_{\ell=1}^{T} \max_{a \in \mathcal{A}} f_\star(x_\ell, a) - f_\star(x_\ell, a_\ell) \leqslant B_{T,\delta'}. \tag{22}$$

For some $B_{T,\delta'} \in \mathbb{R}$. Then policy $\widehat{\pi}_T = \mathrm{Uniform}(\pi_1, \cdot, \pi_T)$ satisfies,

$$\mathbb{E}_{x \sim \mathcal{P}, a \sim \widehat{\pi}_T(x)}[f_\star(x, a)] + \frac{B_{T,\delta'}}{T} + 8B\sqrt{\frac{\log(1/\delta')}{T}} \geqslant \mathbb{E}_{x \sim \mathcal{P}}\left[\max_{a \in \mathcal{A}} f_\star(x, a)\right].$$

with probability at least $1 - 3\delta'$.

*Proof.* Consider the martingale sequences $\{Z_\ell^{(1)}\}_{\ell=1}^{T}$ and $\{Z_\ell^{(2)}\}_{\ell=1}^{T}$ defined as $Z_\ell^{(1)} = \max_{a \in \mathcal{A}} f_\star(x_\ell, a) - \mathbb{E}_{x \sim \mathcal{P}}[\max_{a \in \mathcal{A}} f_\star(x, a)]$ and $Z_\ell^{(2)} = f_\star(x_\ell, a_\ell) - \mathbb{E}_{x \sim \mathcal{P}, a \sim \pi_\ell(x)}[f_\star(x, a)]$ for $\ell \in [T]$.

The bounded function range assumption 3.2 implies $\max(|Z_\ell^{(1)}|, |Z_\ell^{(2)}|) \leqslant 2B$ for all $\ell \leqslant T$. Therefore, Hoeffding Inequality applied to these two martingale sequences (see A.1) and a union bound imply

$$\sum_{\ell=1}^{T} \max_{a \in \mathcal{A}} f_\star(x_\ell, a) + 4B\sqrt{T \log(1/\delta')} \geqslant \sum_{\ell=1}^{T} \mathbb{E}_{x \sim \mathcal{P}}[\max_{a \in \mathcal{A}} f_\star(x, a)] = T\mathbb{E}_{x \sim \mathcal{P}}[\max_{a \in \mathcal{A}} f_\star(x, a)]$$

$$\sum_{\ell=1}^{T} \mathbb{E}_{x \sim \mathcal{P}, a \sim \pi_\ell(x)}[f_\star(x, a)] + 4B\sqrt{T \log(1/\delta')} \geqslant \sum_{\ell=1}^{T} f_\star(x_\ell, a_\ell).$$

simultaneously with probability at least $1 - 2\delta'$. Combining this with inequality 22 and a union bound implies,

$$\sum_{\ell=1}^{T} \mathbb{E}_{x \sim \mathcal{P}, a \sim \pi_\ell(x)}[f_\star(x, a)] + B_{T,\delta'} + 8B\sqrt{T \log(1/\delta')} \geqslant T\mathbb{E}_{x \sim \mathcal{P}}[\max_{a \in \mathcal{A}} f_\star(x, a)].$$

with probability at least $1 - 3\delta'$. Therefore,

$$\mathbb{E}_{x \sim \mathcal{P}, a \sim \widehat{\pi}_T(x)}[f_\star(x, a)] + \frac{B_{T,\delta'}}{T} + 8B\sqrt{\frac{\log(1/\delta')}{T}} \geqslant \mathbb{E}_{x \sim \mathcal{P}}\left[\max_{a \in \mathcal{A}} f_\star(x, a)\right]$$

$\square$

## C Proofs of Section 4

**Lemma 4.6.** *With probability at least* $1 - \frac{\delta}{12}$,

$$\{(f, f') \text{ s.t. } \|f - f'\|_{\widetilde{\mathcal{D}}_t} \leqslant 2\beta_{\mathcal{F}}(t, \delta)\} \subseteq \{(f, f') \text{ s.t. } \|f - f'\|_{\mathcal{D}_t} \leqslant 4\beta_{\mathcal{F}}(t, \delta)\} \tag{6}$$

*Simultaneously for all* $t \in [T]$. *Where* $\{D_t\}_{t=1}^{T}$ *is the dataset sequence resulting of the execution of* EluderPlanner *while* $\{\widetilde{D}_t\}_{t=1}^{T}$ *is the dataset sequence resulting of the execution of the* Sampler *and* $\beta_{\mathcal{F}}(\delta, t)$ *is the confidence radius function defined in Equation 2.*

*Proof.* We will define a process $\{Z_\ell\}_{\ell=1}^{T}$ such that $Z_\ell = (x_\ell, \tilde{x}_\ell, \pi_\ell)$. Here $\tilde{x}_\ell$ is independent and identically distributed to any $x_\ell$ (and in particular to $x_\ell$).

Let $f, f' \in \mathcal{F}$ be two fixed functions. Define two adapted processes $\{f(x_\ell, \pi_\ell(x_\ell)) - f'(x_\ell, \pi_\ell(x_\ell))\}_\ell$ and $\{f(\tilde{x}_\ell, \pi_\ell(\tilde{x}_\ell)) - f'(\tilde{x}_\ell, \pi_\ell(\tilde{x}_\ell))\}_\ell$ (in this definition we have used the fact that $\pi_\ell$ is a sequence of deterministic policies).

Lemma A.5 implies,

$$\sum_{\ell=1}^{t}(f(x_\ell, \pi_\ell(x_\ell)) - f'(x_\ell, \pi_\ell(x_\ell)))^2$$

$$\leqslant \frac{3}{2}\sum_{\ell=1}^{t}\mathbb{E}_{x\sim\mathcal{P}}[(f(x, \pi_\ell(x)) - f'(x, \pi_\ell(x)))^2] + (2C_1^2 + C_2)B^2\log\left(\frac{2t}{\bar{\bar{\delta}}}\right)$$

and

$$\frac{1}{2}\sum_{\ell=1}^{t}\mathbb{E}_{x\sim\mathcal{P}}[(f(x, \pi_\ell(x)) - f'(x, \pi_\ell(x)))^2]$$

$$\leqslant \sum_{\ell=1}^{t}(f(\tilde{x}_\ell, \pi_\ell(\tilde{x}_\ell)) - f'(\tilde{x}_\ell, \pi_\ell(\tilde{x}_\ell)))^2 + (2C_1^2 + C_2)B^2\log\left(\frac{2t}{\bar{\bar{\delta}}}\right)$$

with probability at least $1 - \bar{\delta}$ for all $t \in \mathbb{N}$. Combining these two inequalities we conclude,

$$\sum_{\ell=1}^{t}(f(x_\ell,) - f'(x_\ell))^2 \leqslant 3\sum_{\ell=1}^{t}(f(\tilde{x}_\ell) - f'(\tilde{x}_\ell))^2 + 4(2C_1^2 + C_2)B^2\log\left(\frac{2t}{\bar{\bar{\delta}}}\right)$$

Setting $\bar{\delta} = \delta/|\mathcal{F}|^2$ and a union bound implies that,

$$\|f - f'\|_{\mathcal{D}_t}^2 \leqslant 3\|f - f'\|_{\widehat{\mathcal{D}}_t}^2 + 4(2C_1^2 + C_2)B^2\log\left(\frac{2|\mathcal{F}|t}{\delta}\right)$$

for all $f, f' \in \mathcal{F}$ simultaneously. Since (see Proposition A.7) we define $\beta_\mathcal{F}(\delta, t)$ such that

$$\beta_\mathcal{F}^2(\delta, t) \geqslant 4(2C_1^2 + C_2)B^2\log\left(\frac{2|\mathcal{F}|t}{\delta}\right) \tag{23}$$

we conclude that whenever $f, f' \in \mathcal{F}$ satisfy $\|f - f'\|_{\widetilde{\mathcal{D}}_t} \leqslant 2\beta_\mathcal{F}(t, \delta)$,

$$\|f - f'\|_{\mathcal{D}_t}^2 \leqslant 16\beta_\mathcal{F}^2(t, \delta)$$

and therefore $\|f - f'\|_{\mathcal{D}_t} \leqslant 4\beta_\mathcal{F}(t, \delta)$. This concludes the result. $\qquad\square$

## C.1  Proof of Theorem 4.4

**Theorem 4.4.** *Let $\varepsilon > 0$. There exists a universal constant $c > 0$ such that if*

$$T \geqslant c\frac{\max^2(B, \bar{B}, 1)d_{\mathrm{eluder}}(\mathcal{F}, B/T)\log\left(|\mathcal{F}|T/\delta\right)}{\varepsilon^2} \tag{5}$$

*then with probability at least $1 - \delta$ the eluder planning algorithm's policy $\widehat{\pi}_T$ is $\varepsilon-$optimal .*

In this section we present a more complete version of the proof from Section 4.1. We will actually prove a 'stronger' version of Theorem 4.4.

**Theorem C.1.** *Let $\varepsilon > 0$. There exists a universal constant $c_0 > 0$ such that if*

$$T \geqslant c_0\max\left(\frac{\max^2(B, \bar{B})d_{\mathrm{eluder}}(\mathcal{F}, B/T)\log\left(|\mathcal{F}|T/\delta\right)}{\varepsilon^2}, \frac{Bd_{\mathrm{eluder}}(\mathcal{F}, B/T)}{\varepsilon}\right) \tag{24}$$

*then with probability at least $1 - \delta$ the eluder planning algorithm's policy $\widehat{\pi}_T$ is $\varepsilon-$optimal .*

*Proof.* For readability we have repeated some of the text of the proof in here.

Let $\widehat{f}_t$ as in Equation 3 and define $\mathcal{E}$ as the event where the Standard Least Squares results (see Lemma A.8) hold i.e. $\|\widehat{f}_t - f_\star\|_{\widetilde{\mathcal{D}}_t} \leqslant \beta(t, \delta)$ for all $t \in [T]$ and also Equation 6 from Lemma 4.6 holds for all $t \in [T]$. The results of Lemmas A.8 and 4.6 imply $\mathbb{P}(\mathcal{E}) \geqslant 1 - \frac{\delta}{6}$.

For context $x$ and action $a$ let's denote by $\tilde{f}_t^{x,a}$ as a function achieving the inner maximum in the definition of $\tilde{\pi}_t^{\mathrm{opt}}$ (see Equation 4). When $\mathcal{E}$ holds the policies $\{\tilde{\pi}_t^{\mathrm{opt}}\}_{t=1}^T$ are optimistic in the sense that $\tilde{f}_t^{x,\tilde{\pi}_t^{\mathrm{opt}}(x)}(x, \tilde{\pi}_t^{\mathrm{opt}}(x)) \geqslant \max_{a \in \mathcal{A}} f_\star(x, a)$. To see why this is the case notice that $\mathcal{E}$ holds $f_\star \in \{\|f - \hat{f}_j\|_{\tilde{\mathcal{D}}_t} \leqslant \beta_{\mathcal{F}}(t, \delta)\}$ and therefore

$$\tilde{f}_t^{x,\tilde{\pi}_t^{\mathrm{opt}}(x)}(x, \tilde{\pi}_t^{\mathrm{opt}}(x)) \geqslant \max_{f \in \{\|f - \hat{f}_t\|_{\tilde{\mathcal{D}}_t} \leqslant \beta_{\mathcal{F}}(t, \delta)\}} f(x, \pi_\star(x)) \geqslant f_\star(x, \pi_\star(x)).$$

where $\pi_\star$ corresponds to the optimal policy $\pi_\star(x) = \arg\max_{a \in \mathcal{A}} f_\star(x, a)$. This implies the following sequence of inequalities,

$$\sum_{t=1}^T \max_{a \in \mathcal{A}} f_\star(\tilde{x}_t, a) - f_\star(\tilde{x}_t, \tilde{\pi}_t^{\mathrm{opt}}(\tilde{x}_t)) \leqslant \sum_{t=1}^T \tilde{f}_t^{x,\tilde{\pi}_t^{\mathrm{opt}}(x)}(\tilde{x}_t, \tilde{\pi}_t^{\mathrm{opt}}(\tilde{x}_t)) - f_\star(\tilde{x}_t, \tilde{\pi}_t^{\mathrm{opt}}(\tilde{x}_t))$$

$$\leqslant \sum_{t=1}^T \max_{f, f' \text{ s.t. } \|f - f'\|_{\tilde{\mathcal{D}}_t} \leqslant 2\beta_{\mathcal{F}}(\delta, t)} f(\tilde{x}_t, \tilde{\pi}_t^{\mathrm{opt}}(\tilde{x}_t)) - f'(\tilde{x}_t, \tilde{\pi}_t^{\mathrm{opt}}(\tilde{x}_t))$$

$$\overset{(i)}{\leqslant} \sum_{t=1}^T \max_{f, f' \text{ s.t. } \|f - f'\|_{\mathcal{D}_t} \leqslant 4\beta_{\mathcal{F}}(\delta, t)} f(\tilde{x}_t, \tilde{\pi}_t^{\mathrm{opt}}(\tilde{x}_t)) - f'(\tilde{x}_t, \tilde{\pi}_t^{\mathrm{opt}}(\tilde{x}_t))$$

$$= \sum_{t=1}^T \omega(\tilde{x}_t, \tilde{\pi}_t^{\mathrm{opt}}(\tilde{x}_t), \mathcal{D}_t)$$

$$\leqslant \sum_{t=1}^T \max_{a \in \mathcal{A}} \omega(\tilde{x}_t, a, \mathcal{D}_t)$$

$$= \sum_{t=1}^T \omega(\tilde{x}_t, \pi_t(\tilde{x}_t), \mathcal{D}_t).$$

Where inequality $(i)$ is satisfied when $\mathcal{E}$ holds as a result of Lemma 4.6. Using Hoeffding Inequality (see Lemma A.1) yields $\sum_{t=1}^T \omega(\tilde{x}_t, \pi_t(\tilde{x}_t), \mathcal{D}_t) \leqslant \sum_{t=1}^T \omega(x_t, \pi_t(x_t), \mathcal{D}_t) + 16B\sqrt{T \log\left(\frac{3}{\delta}\right)}$ with probability at least $1 - \delta/3$. Finally Lemma 4.3 implies there exists a universal constant $C > 0$ such that $\sum_{t=1}^T \omega(x_t, \pi_t(x_t), \mathcal{D}_t) \leqslant C \cdot \min\left(BT, Bd_{\mathrm{eluder}}(\mathcal{F}, B/T) + \beta_{\mathcal{F}}(T, \delta)\sqrt{d_{\mathrm{eluder}}(\mathcal{F}, B/T)T}\right)$ and therefore,

$$\sum_{t=1}^T \max_{a \in \mathcal{A}} f_\star(\tilde{x}_t, a) - f_\star(\tilde{x}_t, \tilde{\pi}_t^{\mathrm{opt}}(\tilde{x}_t)) \leqslant C \cdot \min\left(BT, Bd_{\mathrm{eluder}}(\mathcal{F}, B/T) + \beta_{\mathcal{F}}(T, \delta)\sqrt{d_{\mathrm{eluder}}(\mathcal{F}, B/T)T}\right).$$

With probability at least $1 - \delta$. The Online to Batch Conversion Lemma B.1 implies there exists a universal constant $C > 0$ such that,

$$\mathbb{E}_{x \sim \mathcal{P}, a \sim \hat{\pi}_T(x)}\left[f_\star(x, a)\right] +$$
$$\frac{C \cdot \min\left(BT, Bd_{\mathrm{eluder}}(\mathcal{F}, B/T) + \sqrt{d_{\mathrm{eluder}}(\mathcal{F}, B/T)\beta_{\mathcal{F}}(T, \delta)T}\right)}{T} + 8B\sqrt{\frac{\log(1/\delta)}{T}}$$
$$\geqslant \max_\pi \mathbb{E}_{x \sim \mathcal{P}}\left[f_\star(x, \pi(x))\right].$$

with probability at least $1 - 3\delta$ where $\hat{\pi}_T = \mathrm{Uniform}(\pi_1^{\mathrm{opt}}, \cdots, \pi_T^{\mathrm{opt}})$. Finally setting $T$ sufficiently large such that,

$$T \geqslant \frac{4CB}{\varepsilon}$$
$$T \geqslant \frac{4Bd_{\mathrm{eluder}}(\mathcal{F}, B/T)}{\varepsilon}$$
$$T \geqslant \frac{16d_{\mathrm{eluder}}(\mathcal{F}, B/T)\beta_{\mathcal{F}}^2(T, \delta)}{\varepsilon^2}$$
$$T \geqslant \frac{256B \log(1/\delta)}{\varepsilon^2}$$

we conclude there exists a universal constant $c_0 > 0$ such that if

$$T \geqslant c_0 \max\left(\frac{\max^2(B, \bar{B})d_{\mathrm{eluder}}(\mathcal{F}, B/T)\log\left(|\mathcal{F}|T/\delta\right)}{\varepsilon^2}, \frac{Bd_{\mathrm{eluder}}(\mathcal{F}, B/T)}{\varepsilon}\right)$$

then with probability at least $1 - \delta$,

$$\mathbb{E}_{x\sim\mathcal{P}}\left[f_\star(x, \widehat{\pi}_T(x))\right] + \varepsilon \geqslant \max_\pi \mathbb{E}_{x\sim\mathcal{P}}\left[f_\star(x, \pi(x))\right].$$

$\square$

We now return to the proof of Theorem 4.4. We can simplify inequality 24 by instead imposing the condition,

$$T \geqslant c\frac{\max^2(B, \bar{B}, 1)d_{\mathrm{eluder}}(\mathcal{F}, B/T)\log\left(|\mathcal{F}|T/\delta\right)}{\varepsilon^2}$$

for some universal constant $c > 0$. This finalizes the proof.

**Comparison with Linear Experiment Planning** . Using standard techniques the results of Theorem 4.4 can be adapted to the case when the function class $\mathcal{F}$ is infinite but admits a metrization and has a covering number. In this case the sample complexity will scale not with $\log(|\mathcal{F}|)$ but instead with the logarithm of the covering number of $\mathcal{F}$. For example, in the case of linear functions, the logarithm of the covering number of $\mathcal{F}$ under the $\ell_2$ norm will scale as $d$, the ambient dimension of the space up to logarithmic factors of $T$ (see for example Lemma D.1 of [14]). Plugging this into the sample guarantees of Theorem 4.4 recovers the $\widetilde{\mathcal{O}}\left(d^2/\varepsilon^2\right)$ sample complexity for linear experiment planning from [53].

# D    Proof of Theorem 5.1

**Theorem 5.1.** *There exists a universal constant $\tilde{c} > 0$ such that if $T \geqslant \tilde{c}\frac{\max^2(B,\bar{B})|\mathcal{A}|\log(|\mathcal{F}|/\varepsilon\delta)}{\varepsilon^2}$ then with probability at least $1 - \delta$ the uniform planning algorithm's policy $\widehat{\pi}_T$ is $\varepsilon-$optimal.*

*Proof.* Due to the realizability assumption ($r_t = f_\star(x_t, a_t) + \xi_t$) the instantaneous regret of $\widehat{\pi}_T$ on context $x \in \mathcal{X}$ equals $\max_a f_\star(x, a) - f_\star(x, \widehat{\pi}_T(x))$. Let $\pi_\star(x) = \arg\max_{a\in\mathcal{A}} f_\star(x, a)$. The following inequalities hold,

$$\max_{a\in\mathcal{A}} f_\star(x, a) - f_\star(x, \widehat{\pi}_T(x)) = f_\star(x, \pi_\star(x)) - f_\star(x, \widehat{\pi}_T(x))$$

$$\overset{(i)}{\leqslant} f_\star(x, \pi_\star(x)) - \widehat{f}_T(x, \pi_\star(x)) + \widehat{f}_T(x, \widehat{\pi}_T(x)) - f_\star(x, \widehat{\pi}_T(x))$$

$$\leqslant 2\max_{a\in\mathcal{A}}|f_\star(x, a) - \widehat{f}_T(x, a)|$$

Where $(i)$ holds because $\widehat{f}_T(x, \pi_\star(x)) \leqslant \widehat{f}_T(x, \widehat{\pi}_T(x))$. Therefore,

$$\mathbb{E}_{x\sim\mathcal{P}}\left[\max_{a\in\mathcal{A}} f_\star(x, a) - f_\star(x, \widehat{\pi}_T(x))\right] \leqslant 2\mathbb{E}_{x\sim\mathcal{P}}\left[\max_{a\in\mathcal{A}}|f_\star(x, a) - \widehat{f}_T(x, a)|\right]$$

$$\overset{(i)}{\leqslant} 2\sqrt{\mathbb{E}_{x\sim\mathcal{P}}\left[\max_{a\in\mathcal{A}}\left(f_\star(x, a) - \widehat{f}_T(x, a)\right)^2\right]}$$

$$\overset{(ii)}{\leqslant} 2\sqrt{|\mathcal{A}|\mathbb{E}_{x\sim\mathcal{P}, a\sim\mathrm{Uniform}(\mathcal{A})}\left[\left(f_\star(x, a) - \widehat{f}_T(x, a)\right)^2\right]}.$$

$$(25)$$

Inequality $(i)$ holds because of Jensen's inequality. Inequality $(ii)$ follows because $\max_{a\in\mathcal{A}}\left(f_\star(x, a) - \widehat{f}_T(x, a)\right)^2 \leqslant \sum_{a\in\mathcal{A}}\left(f_\star(x, a) - \widehat{f}_T(x, a)\right)^2$.

Finally the context generating distributions equal $\mathcal{P}$ at all timesteps and the policies equal $\mathrm{Uniform}(\mathcal{A})$ standard least squares analysis (Lemma A.8) yields that with probability at least $1 - \delta$

$$\mathbb{E}_{x\sim\mathcal{P}, a\sim\mathrm{Uniform}(\mathcal{A})}\left[\left(f_\star(x, a) - \widehat{f}_T(x, a)\right)^2\right] \leqslant \frac{C_{T,\delta}}{T}.$$

Therefore we conclude that,

$$\mathbb{E}_{x \sim \mathcal{P}} \left[ \max_{a \in \mathcal{A}} f_\star(x, a) - f_\star(x, \widehat{\pi}_T(x)) \right] \leqslant 2\sqrt{\frac{|\mathcal{A}| C_{T,\delta}}{T}}$$

Since $C_{T,\delta} = c'(B^2 + \bar{B}^2) \log\left(\frac{|\mathcal{F}|t}{\delta}\right)$ for some universal constant $c' > 0$ we conclude that,

$$\mathbb{E}_{x \sim \mathcal{P}} \left[ \max_{a \in \mathcal{A}} f_\star(x, a) - f_\star(x, \widehat{\pi}_T(x)) \right] \leqslant 2\sqrt{\frac{c'|\mathcal{A}| \max^2(B, \bar{B}) \log\left(\frac{|\mathcal{F}|t}{\delta}\right)}{T}}$$

Setting $g(T) = 4\frac{c'|\mathcal{A}| \max^2(B,\bar{B}) \log\left(\frac{|\mathcal{F}|t}{\delta}\right)}{T}$ in Lemma A.6 implies there exists a universal constant $\tilde{c} > 0$ such that $g(T) \leqslant \varepsilon^2$ for all $T \geqslant \tilde{c}\frac{\max^2(B,\bar{B})|\mathcal{A}| \log\left(\frac{|\mathcal{F}|}{\varepsilon\delta}\right)}{\varepsilon^2}$. The result follows.

$\square$

## D.1 Proof of Theorem 6.1

In order to prove Theorem 6.1 we will assume a gaussian model for the noise variables $\xi_t \sim \mathcal{N}(0, 1)$. This will allow us to use the appropriate information theoretic tools.

**Theorem 6.1.** *Let $\varepsilon > 0$, $T \in \mathbb{N}$. Consider the action set $\mathcal{A}_{\text{tree}}$ and function class $\mathcal{F}_{\text{tree}}$ and a reward noise process such that $\xi_t \sim \mathcal{N}(0,1)$ conditionally for all $t \in [T]$. For any planning algorithm* Alg *there is a function $f_\star \in \mathcal{F}_{\text{tree}}$ such that when $T \leqslant \frac{2^{L-5}}{9\varepsilon^2}$ and* Alg *interacts with $f_\star$ then,*

$$\mathbb{E}_{\text{Alg}, f_\star} \left[ \mathbb{E}_{a \sim \widehat{\pi}_T} \left[ f_\star(a) \right] \right] < \max_{a \in \mathcal{A}_{\text{tree}}} f_\star(a) - \varepsilon.$$

*Moreover, there is an adaptive algorithm* Alg$_{\text{adaptive}}$ *such that if $T \geqslant \frac{2L \log(2L/\varepsilon)}{\varepsilon^2}$,*

$$\mathbb{E}_{\text{Alg}_{\text{adaptive}}} \left[ \mathbb{E}_{a \sim \widehat{\pi}_T} \left[ f(a) \right] \right] \geqslant \max_{a \in \mathcal{A}_{\text{tree}}} f(a) - \varepsilon.$$

*for all $f \in \mathcal{F}_{\text{tree}}$. Where $\mathbb{E}_{\widetilde{\text{Alg}}, \widetilde{f}} \left[ \cdot \right]$ is the expectation over the randomness of $\widetilde{\text{Alg}}$ and the environment for target function $\widetilde{f}$, and $\widehat{\pi}_T$ is the algorithm's final policy guess after the sampling phase.*

*Proof.* We use the notation $a_1, \cdots, a_T$ to denote a (random) sample path of query tuples for algorithm Alg with observed responses $r_1, \cdots, r_T$ where $r_t = f(a_t) + \xi_t$. We'll assume that $\xi_t \sim \mathcal{N}(0, 1)$.

Algorithm Alg interacts for $T$ timesteps during the planning phase with $\mathcal{A}_{\text{tree}}$ and produces a sequence of policies $\{\pi_t\}_{t=1}^T$ to deploy during the sampling phase. The data collected from these deployment policies is then used by the algorithm to define a candidate $\varepsilon$-optimal policy $\widehat{\pi}_T$.

Since we are studying the context-less setting, the underlying context distribution equals a delta mass over the empty context and therefore the distribution over the sampling policy sequence $\{\pi_t\}_{t=1}^T$ is independent of $f$. Let $n_T(a)$ be the random variable specifying how many times action $a$ was pulled during the sampling phase of algorithm Alg. This random variable is independent of $f$.

Let $\widetilde{a}$ and $\widetilde{b}$ be two consecutive leaf actions (i.e. inhabiting a size 2 sub-tree of $\mathcal{A}_{\text{tree}}$) such that,

$$\mathbb{E}_{\text{Alg}}[n_T(\widetilde{a}) + n_T(\widetilde{b})] \leqslant \frac{T}{2^{L-2}}.$$

Let $\mathbf{p}_{\widetilde{a}}$ be the path leading to $\widetilde{a}$ and $\mathbf{p}_{\widetilde{b}}$ be the path leading to $\widetilde{b}$. We'll use the notation $\mathbb{P}_a^{\mathbf{P}_{\widetilde{a}}}, \mathbb{P}_a^{\mathbf{P}_{\widetilde{b}}}$ to denote the distribution of arm $a$ in the world specified by $f^{\mathbf{P}_{\widetilde{a}}}$ and $f^{\mathbf{P}_{\widetilde{b}}}$ respectively. The divergence decomposition for markov processes (see for example Lemma 15.1 in [28] ) implies

$$\text{KL}(\mathbb{P}_{\text{Alg}, f^{(\mathbf{P}_{\widetilde{a}})}} \parallel \mathbb{P}_{\text{Alg}, f^{(\mathbf{P}_{\widetilde{b}})}}) = \sum_{a \in \mathcal{A}_{\text{tree}}} \mathbb{E}_{\text{Alg}} \left[ n_T(a) \right] \text{KL}(\mathbb{P}_a^{\mathbf{P}_{\widetilde{a}}} \parallel \mathbb{P}_a^{\mathbf{P}_{\widetilde{b}}})$$

We have used the notation $\mathbb{E}_{\text{Alg}} \left[ \cdot \right]$ above because the $n_T(a)$ random variables do not depend on $f$. Notice that for all $a \notin \{\widetilde{a}, \widetilde{b}\}$, $\text{KL}(\mathbb{P}_a^{\mathbf{P}_{\widetilde{a}}} \parallel \mathbb{P}_a^{\mathbf{P}_{\widetilde{b}}}) = 0$. In contrast $\text{KL}(\mathbb{P}_a^{\mathbf{P}_{\widetilde{a}}} \parallel \mathbb{P}_a^{\mathbf{P}_{\widetilde{b}}}) = 72\varepsilon^2$ for $a \in \{\widetilde{a}, \widetilde{b}\}$. Plugging this into the equation above,

$$\text{KL}(\mathbb{P}_{\text{Alg}, f^{(\mathbf{P}_{\widetilde{a}})}} \parallel \mathbb{P}_{\text{Alg}, f^{(\mathbf{P}_{\widetilde{b}})}}) = 72\mathbb{E}_{\text{Alg}} \left[ n_T(\widetilde{a}) + n_T(\widetilde{b}) \right] \varepsilon^2 \leqslant \frac{\varepsilon^2 T}{2^{L-3}}.$$

Define the event $\mathcal{E} = \{\widehat{\pi}_T(\widetilde{a}) < \widehat{\pi}_T(\widetilde{b})\}$. The Huber-Bretagnolle inequality implies,

$$\mathbb{P}_{\mathrm{Alg},f^{(\mathbf{P}_{\widetilde{a}})}}\left(\mathcal{E}\right) + \mathbb{P}_{\mathrm{Alg},f^{(\mathbf{P}_{\widetilde{b}})}}\left(\mathcal{E}^c\right) \geqslant \exp\left(-\mathrm{KL}(\mathbb{P}_{\mathrm{Alg},f^{(\mathbf{P}_{\widetilde{a}})}} \parallel \mathbb{P}_{\mathrm{Alg},f^{(\mathbf{P}_{\widetilde{b}})}})\right) \geqslant \exp\left(-\frac{9\varepsilon^2 T}{2^{L-5}}\right). \quad (26)$$

Notice that when $\mathcal{E}$ holds, $2\widehat{\pi}_T(\widetilde{a}) \leqslant \widehat{\pi}_T(\widetilde{a}) + \widehat{\pi}_T(\widetilde{b}) \leqslant 1$ and therefore $\widehat{\pi}_T(\widetilde{a}) \leqslant \frac{1}{2}$. Thus if $\mathcal{E}$ holds,

$$\mathbb{E}_{\mathrm{Alg},f^{(\mathbf{P}_{\widetilde{a}})}}\left[\mathbb{E}_{a\sim\widehat{\pi}_T}\left[f_\star(a)\right]\Big|\mathcal{E}\right] \leqslant \frac{1}{2} + \frac{1}{2}\left(1 - 12\varepsilon\right) = 1 - 6\varepsilon. \quad (27)$$

In this case,

$$\mathbb{E}_{\mathrm{Alg},f^{(\mathbf{P}_{\widetilde{a}})}}\left[\mathbb{E}_{a\sim\widehat{\pi}_T}\left[f_\star(a)\right]\right] \leqslant \mathbb{E}_{\mathrm{Alg},f^{(\mathbf{P}_{\widetilde{a}})}}\left[\mathbb{E}_{a\sim\widehat{\pi}_T}\left[f_\star(a)\right]\Big|\mathcal{E}\right]\mathbb{P}_{\mathrm{Alg},f^{(\mathbf{P}_{\widetilde{a}})}}\left(\mathcal{E}\right) +$$

$$\mathbb{E}_{\mathrm{Alg},f^{(\mathbf{P}_{\widetilde{a}})}}\left[\mathbb{E}_{a\sim\widehat{\pi}_T}\left[f_\star(a)\right]\Big|\mathcal{E}^c\right]\mathbb{P}_{\mathrm{Alg},f^{(\mathbf{P}_{\widetilde{a}})}}\left(\mathcal{E}^c\right)$$

$$\leqslant \mathbb{E}_{\mathrm{Alg},f^{(\mathbf{P}_{\widetilde{a}})}}\left[\mathbb{E}_{a\sim\widehat{\pi}_T}\left[f_\star(a)\right]\Big|\mathcal{E}\right]\mathbb{P}_{\mathrm{Alg},f^{(\mathbf{P}_{\widetilde{a}})}}\left(\mathcal{E}\right) + \mathbb{P}_{\mathrm{Alg},f^{(\mathbf{P}_{\widetilde{a}})}}\left(\mathcal{E}^c\right)$$

$$\overset{(i)}{\leqslant} 1 - 6\varepsilon\mathbb{P}_{\mathrm{Alg},f^{(\mathbf{P}_{\widetilde{a}})}}\left(\mathcal{E}\right) \quad (28)$$

Where inequality $(i)$ holds because of Equation 27. Similarly we can infer that,

$$\mathbb{E}_{\mathrm{Alg},f^{(\mathbf{P}_{\widetilde{b}})}}\left[\mathbb{E}_{a\sim\widehat{\pi}_T}\left[f_\star(a)\right]\right] \leqslant 1 - \varepsilon\mathbb{P}_{\mathrm{Alg},f^{(\mathbf{P}_{\widetilde{b}})}}\left(\mathcal{E}\right). \quad (29)$$

Finally, combining Equations 28 and 29 with the result of Huber-Bretagnolle (see Equation 26) implies,

$$\mathbb{E}_{\mathrm{Alg},f^{(\mathbf{P}_{\widetilde{a}})}}\left[\mathbb{E}_{a\sim\widehat{\pi}_T}\left[f_\star(a)\right]\right] + \mathbb{E}_{\mathrm{Alg},f^{(\mathbf{P}_{\widetilde{b}})}}\left[\mathbb{E}_{a\sim\widehat{\pi}_T}\left[f_\star(a)\right]\right] \leqslant 2 - 6\varepsilon\exp\left(-\frac{9\varepsilon^2 T}{2^{L-5}}\right).$$

Hence, as long as $T \leqslant \frac{2^{L-5}}{9\varepsilon^2}$, then $\exp\left(-\frac{9\varepsilon^2 T}{2^{L-5}}\right) \geqslant \exp(-1)$ and therefore $\mathbb{E}_{\mathrm{Alg},f^{(\mathbf{P}_{\widetilde{a}})}}\left[\mathbb{E}_{a\sim\widehat{\pi}_T}\left[f_\star(a)\right]\right] + \mathbb{E}_{\mathrm{Alg},f^{(\mathbf{P}_{\widetilde{b}})}}\left[\mathbb{E}_{a\sim\widehat{\pi}_T}\left[f_\star(a)\right]\right] \leqslant 2 - 6\varepsilon\exp(-1)$. And therefore at least one $\widetilde{c} \in \{\widetilde{a},\widetilde{b}\}$ satisfies

$$\mathbb{E}_{\mathrm{Alg},f^{(\mathbf{P}_{\widetilde{c}})}}\left[\mathbb{E}_{a\sim\widehat{\pi}_T}\left[f_\star(a)\right]\right] \leqslant 1 - \frac{3\varepsilon}{e} < 1 - \varepsilon.$$

This finalizes the first result.

For the second result we analyze the following algorithm,

---
**Algorithm 3** Adaptive Tree Sampling

1: Input: Number of samples per node $M$.
2: Initialize current node as $a_{\mathrm{curr}} = a_{1,1}$
3: **for** $j = 1\ldots L-1$ **do**
4:     Expand current node $(b,c) = \mathrm{Children}(a_{\mathrm{curr}})$.
5:     Collect $M$ samples from each node $b$ and $c$ and compute mean rewards $\hat{r}_b$ and $\hat{r}_c$.
6:     Update $a_{\mathrm{curr}} = \arg\max_{e\in(b,c)}\hat{r}_e$.
7: **end for**
8: **Output** $a_{\mathrm{output}} = a_{\mathrm{curr}}$.

---

Let's assume $f^{(\mathbf{P})}$ is the world's descriptor for some path $\mathbf{p}$. We will analyze the performance of Algorithm 3 where $M = \frac{\log(2L/\delta)}{\varepsilon^2}$. The output of this algorithm will be a policy centered around the output action $a$.

Let's first observe that Hoeffding inequality (Lemma A.1) and the union bound imply that $|\hat{r}_b - f^{(\mathbf{P})}(b)| \leqslant \varepsilon$, with probability at least $1 - \frac{\delta}{L}$ and $|\hat{r}_e - f^{(\mathbf{P})}(e)| \leqslant \varepsilon$ for $e \in \{b,c\}$ simultaneously. Let's call this event $\mathcal{E}_a$.

Lets assume $a_{\mathrm{curr}} \in \mathbf{p}$ and is not a leaf node, and w.l.o.g. when $b \in \mathbf{p}$ and $c \notin \mathbf{p}$, such that $f^{(\mathbf{P})}(b) \geqslant 1 - 2\varepsilon$ and $f^{(\mathbf{P})}(c) = 1 - 12\varepsilon$. If $\mathcal{E}_a$ holds,

$$\hat{r}_b \geqslant f^{(\mathbf{P})}(b) - \varepsilon \geqslant 1 - 3\varepsilon > 1 - 11\varepsilon \geqslant f^{(\mathbf{P})}(c) + \varepsilon \geqslant \hat{r}_c.$$

We conclude that in this case $\hat{r}_b > \hat{r}_c$ and therefore the updated node $a$ (Line 6 of Algorithm 3) will also be in $\mathbf{p}$.

Combining these results with union bounds applied conditionally over all $a_{\text{curr}}$ in Algorithm 3's sample path, we conclude that with probability at least $1 - \delta$, the output action $a_{\text{output}}$ will equal the last action in $\mathbf{p}$'s path, and therefore the optimizer of $f^{(\mathbf{p})}$. Denote this event as $\mathcal{E}_\delta$.

If we denote this algorithm by $\text{Alg}_{\text{adaptive}}$, define $T \geqslant 2(L-1)M$ and $\widehat{\pi}_T = \delta_{a_{\text{output}}}$ where $a_{\text{output}}$ is the output action, we obtain

$$\mathbb{E}_{\text{Alg}_{\text{adaptive}}}\left[\mathbb{E}_{a \sim \widehat{\pi}_T}[f(a)]\right] \geqslant \mathbb{E}_{\text{Alg}_{\text{adaptive}}}\left[\mathbb{E}_{a \sim \widehat{\pi}_T}[f(a)]\Big|\mathcal{E}_\delta\right]\mathbb{P}_{\text{Alg}_{\text{adaptive}}}(\mathcal{E}_\delta) \geqslant 1 - \delta$$

Thus, setting $\delta = \varepsilon$ and therefore for all $T \geqslant \frac{2L\log(2L/\varepsilon)}{\varepsilon^2}$ we obtain the desired result. $\qquad\square$

## D.2 Eluder dimension of $\mathcal{F}_{\text{tree}}$

**Lemma D.1.** *The eluder dimension of the function class $\mathcal{F}_{\text{tree}}$ satisfies,*

$$d_{\text{eluder}}(\mathcal{F}_{\text{tree}}, \varepsilon) \geqslant 2^{L-1} - 1.$$

*Proof.* To prove this statement it is sufficient to exhibit an $\varepsilon-$independent sequence of actions that certifies the eluder dimension is large. To do this let's we use the ordered list of leaf actions. For simplicity we will call $f_i$ to the function defined by the path ending at leaf node $a_{L,i}$. The sequence

Consider the sequence of actions $a_{L,1}, \cdots, a_{L,2^{L-1}-2}$. Notice that for all $n \in [2^{L-1} - 2]$ the two functions $f_{n+1}$ and $f_{2^{L-1}}$ agree on $a_{L,1}, \cdots, a_{L,n}$. Thus for all $n \in [2^{L-1} - 2]$,

$$\sum_{i=1}^{n}(f_{n+1}(a_i) - f_{2^{L-1}}(a_i)) = 0,$$

while $f_{n+1}(a_{n+1}) - f_{2^{L-1}}(a_{n+1}) = 12\varepsilon > \varepsilon$. This shows $a_{L,1}, \cdots, a_{L,2^{L-1}-1}$ is a size $2^{L-1} - 1$ independent sequence. Since the eluder dimension equals the size of the largest independence sequence this implies the result. $\qquad\square$

# E Model Selection

## E.1 Proof of Proposition 7.1

**Proposition 7.1.** *There exists a universal constant $\underline{c} > 0$ such that,*

$$\mathbb{E}_{x \sim \mathcal{P}, a \sim \text{Uniform}(\mathcal{A})}\left[\left(f_\star(x,a) - \widehat{f}_T^{(i)}(x,a)\right)^2\right] \leqslant \frac{\underline{c}\max^2(B, \bar{B})\log(T\max(M, |\mathcal{F}_{i_\star}|)/\delta)}{T}.$$

*With probability at lest $1 - \delta$.*

*Proof.* Let's start by considering $\widehat{f}_T^{(i_\star)}$. Standard analysis (Lemma A.8) yields that with probability at least $1 - \delta/2$

$$\mathbb{E}_{x \sim \mathcal{P}, a \sim \text{Uniform}(\mathcal{A})}\left[\left(f_\star(x,a) - \widehat{f}_T^{(i_\star)}(x,a)\right)^2\right] \leqslant \frac{2C_{T,\delta/2}^{(i_\star)}}{T}. \tag{30}$$

Let's write $r_\ell = f_\star(x_\ell, a_\ell) + \xi_\ell$ where $\xi_\ell$ are $1-$subgaussian conditionally zero mean. Lemma A.9 yields that,

$$\mathbb{E}_{x \sim \mathcal{P}, a \sim \text{Uniform}(\mathcal{A})}\left[\left(f_\star(x,a) - \widehat{f}_T^{(i)}(x,a)\right)^2\right]$$

$$\leqslant \frac{8}{T}\sum_{(x_\ell, a_\ell, r_\ell) \in \widetilde{\mathcal{D}}_T^{(\text{Test})}}\left(\widehat{f}_T^{(i)}(x_\ell, a_\ell) - r_\ell\right)^2 - \frac{8}{T}\sum_{\ell=1}^{T/2}\xi_\ell^2 + \frac{\bar{C}'(\bar{B}^2 + B^2)\log(TM/\delta)}{T} \tag{31}$$

for all $i \in [M]$ simultaneously (union bound) with probability at least $1 - \delta/2$ for some constant $\bar{\underline{C}}' > 0$. Incorporating the definition of $\underline{i}$ as the minimizer of the empirical loss (see Equation 7) into Equation 31 implies

$$\mathbb{E}_{x \sim \mathcal{P}, a \sim \text{Uniform}(\mathcal{A})} \left[ \left( f_\star(x, a) - \widehat{f}_T^{(\underline{i})}(x, a) \right)^2 \right]$$

$$\leqslant \frac{8}{T} \sum_{(x_\ell, a_\ell, r_\ell) \in \widetilde{\mathcal{D}}_T^{(\text{Test})}} \left( \widehat{f}_T^{(\underline{i})}(x_\ell, a_\ell) - r_\ell \right)^2 - \frac{8}{T} \sum_{\ell=1}^{T/2} \xi_\ell^2 + \frac{\bar{\underline{C}}'(\bar{B}^2 + B^2) \log(TM/\delta)}{T}$$

$$\leqslant \frac{8}{T} \sum_{(x_\ell, a_\ell, r_\ell) \in \widetilde{\mathcal{D}}_T^{(\text{Test})}} \left( \widehat{f}_T^{(i_\star)}(x_\ell, a_\ell) - r_\ell \right)^2 - \frac{8}{T} \sum_{\ell=1}^{T/2} \xi_\ell^2 + \frac{\bar{\underline{C}}'(\bar{B}^2 + B^2) \log(TM/\delta)}{T}$$

with probability at least $1 - \delta/2$ where $\underline{i}$ is on the LHS and $i_\star$ is on the right hand side. Using the RHS of the inequality from Lemma A.9 yields that,

$$\mathbb{E}_{x \sim \mathcal{P}, a \sim \text{Uniform}(\mathcal{A})} \left[ \left( f_\star(x, a) - \widehat{f}_T^{(\underline{i})}(x, a) \right)^2 \right]$$

$$\leqslant 16 \mathbb{E}_{x \sim \mathcal{P}, a \sim \text{Uniform}(\mathcal{A})} \left[ \left( f_\star(x, a) - \widehat{f}_T^{(i_\star)}(x, a) \right)^2 \right] + \frac{\bar{\underline{C}}''(\bar{B}^2 + B^2) \log(TM/\delta)}{T}$$

$$\tag{32}$$

for some universal constant $\bar{\underline{C}}'' > 0$ with probability at least $1 - 3\delta/4$. Finally, the Least Squares guarantee of Lemma A.8 applied to $\mathcal{F}_{i_\star}$ and least squares estimator $\widehat{f}_T^{(i_\star)}$ (where all $\mathcal{P}_t = \mathcal{P}$) implies the first term of the RHS of the equation above can be bounded as,

$$\mathbb{E}_{x \sim \mathcal{P}, a \sim \text{Uniform}(\mathcal{A})} \left[ \left( f_\star(x, a) - \widehat{f}_T^{(i_\star)}(x, a) \right)^2 \right] \leqslant \frac{2C_{T/2, \delta/4}}{T}.$$

with probability at least $1 - \frac{\delta}{4}$. Since in this case $C_{T/2, \delta/4} = \mathcal{O}\left( (B^2 + \bar{B}^2) \log\left( \frac{T |\mathcal{F}_{i_\star}|}{\delta} \right) \right)$, combining this with Equation 32 and a union bound implies

$$\mathbb{E}_{x \sim \mathcal{P}, a \sim \text{Uniform}(\mathcal{A})} \left[ \left( f_\star(x, a) - \widehat{f}_T^{(\underline{i})}(x, a) \right)^2 \right] \leqslant \frac{\underline{c}(\bar{B}^2 + B^2) \log(T \max(M, |\mathcal{F}_{i_\star}|)/\delta)}{T}.$$

for some universal constant $\underline{c} > 0$ and with probability at least $1 - \delta$. The result follows. $\qquad \square$