# OpenReview forum: "Experiment Planning with Function Approximation"
_NeurIPS.cc/2023/Conference — NeurIPS 2023 poster_

### Official Review · Reviewer_xUKA · 2023-07-10

**Soundness:** 3 good
**Presentation:** 4 excellent
**Contribution:** 3 good
**Rating:** 7
**Confidence:** 3

**Summary:**

The paper studies the problem of static experiment planning under the function approximation setting, which contrasts with the adaptive setting where the reward is not observable during the static planning phase. The paper proposes two static planning strategies: (1) EluderPlanner based on Eluder Dimension, and (2) Naive uniform sampler, and analyzes their regret, achieving a cumulative regret similar to the OLS and SquareCB algorithms in their adaptive counterparts.

The paper takes a step further by designing a special tree-structured function class that highlights the gap between the proposed static planning algorithms and the lower bound of adaptive learning in terms of regret.

Finally, the paper addresses the problem of model selection and demonstrates the possibility of constructing an $\varepsilon$-optimal policy.

**Strengths:**

Clear motivation and good writing:
- I enjoyed reading the motivational example in the introduction that explains why adaptive learning scenarios can be challenging.
- All the problem setups and the relations with the related literature are clearly explained.
- The structure of the paper is good and easy to follow.
- Assumptions are well explained, and the definition of Eluder dimension is helpful.

Novelty:
- Combining Eluder dimension and static experiment planning appears to be a novel approach.
- The lower bound construction in Section 5 is informative and demonstrates a fundamental gap between static planning and adaptive learning.

Theoretical soundness:
- I read through the main text, and all the derivations seem correct to me.

**Weaknesses:**

Related work on reward-free RL: I found reward-free RL to be closely related, especially [1], which also describes reward-free RL under function approximation. It would be good to have a short paragraph discussing reward-free RL in general.

[1] Shuang Qiu, Jieping Ye, Zhaoran Wang, Zhuoran Yang. "On Reward-Free RL with Kernel and Neural Function Approximations: Single-Agent MDP and Markov Game."

**Questions:**

I am not familiar with Eluder dimension, but I find the construction of the function class in Section 5 extremely interesting. Will there be a similar gap for some of our familiar non-linear function classes, such as neural networks or kernel spaces, between static planning and adaptive learning?

---

> ### Author Rebuttal · Authors · 2023-08-08
>
> We want to thank the reviewer for their kind comments. A more in-depth discussion of the relationship between experiment planning and Reward-free RL is a great idea to improve our manuscript. We will use the extra page granted at camera ready time to achieve this. We will add the citation the reviewer has mentioned to the final version of our work.
>
> “Will there be a similar gap for some of our familiar non-linear function classes”. This is an extremely interesting question! We believe the conditional tree structured class can be embedded as a subclass of a linear or more complex kernel or NN class which would address this question. It is certainly a very interesting avenue for future research.

---

### Official Review · Reviewer_TWux · 2023-07-11

**Soundness:** 3 good
**Presentation:** 3 good
**Contribution:** 3 good
**Rating:** 6
**Confidence:** 2

**Summary:**

The paper addresses the problem of experiment planning with function approximation in contextual bandit problems.
It is intended to solve the scenario that the datasets include a large amount of contexts while no rewards.
The authors propose two experiment planning strategies that are compatible with function approximation: an eluder planning and sampling procedure. The theoretical results guarantee that the policy converges to optimially under the time constrained by the eluder dimension.

**Strengths:**

This algorithm achieves in this setting: the reward signal is not abundant while there are a large amount of the contexts. The uniform experiment planning algorithms help solve the setting.

Two important questions are answered in this problem:

(1)are existing adaptive cumulative regret algorithms already optimal for simple regret minimization?

(2) is static experiment planning sufficient to match the simple regret guarantees of adaptive learning for realizable contextual bandits?


**Weaknesses:**

1.The paper does not provide a detailed comparison with existing experiment planning algorithms to show the improvement of the algorithm, especially in the function approximation like linear function. Enough related works and comparisons are needed in this paper.

2.The main results and the proof is lack of proof intuition to highlight the keypoint. The results need a clear and intuitive proof sketch to help readers follow the idea.
The results are lack of a lower bound to guarantee the optimality of the algorithm. A tight bound is needed. If not, the gap of the lower bound and the upper bound is needed to state.

3.The paper could benefit from a more detailed discussion of the limitations of the proposed approach.

**Questions:**

1. Is there a specific experiment to show the performance of the algorithm?

**Limitations:**

No. The paper could benefit from a more detailed discussion of the limitations of the proposed approach.

---

> ### Author Rebuttal · Authors · 2023-08-08
>
> We want to thank the reviewer for their comments. We would like to start by pushing back on the reviewers comment regarding the comparison with existing experiment planning algorithms (see Weaknesses issue 1). The setting of experiment planning was introduced first by [37]. The only existing results we are aware of are the linear algorithm of [37]. As we have explained in our manuscript, our eluder Planner and sampler algorithms recover and subsume the bounds of [37]. This is because the eluder dimension of a linear class agrees with the underlying dimension of the space. We will certainly add more explanation regarding this to the final version of the paper.
>
> “The main results and the proof is lack of proof intuition“ Thanks so much for pointing out useful suggestions on how to make our paper easier to read and understand by readers. We will make use of the extra page to make our explanations clearer to the reader.
>
> “The results are lack of a lower bound … “ we are somewhat baffled by this comment. Deriving lower bounds for the general function approximation regime is an extremely complex question. For example, such bounds are not fully understood for the setting of adaptive learning algorithms (bandits and RL), a vastly more developed area of research than the much more recently developed setting of experiment planning. We hope these has persuaded the reviewer that matching lower and upper bounds for the function approximation regime in experiment planning is an extremely unrealistic measuring rod and beyond the reasonable scope of a neurips submission.

---

> > ### Comment · Reviewer_TWux · 2023-08-14
> > **5 to 6**
> >
> > Thank you very much for your detailed response! My concern has been well-addressed and thus I would like to increase my score.

---

### Official Review · Reviewer_ESHY · 2023-07-11

**Soundness:** 1 poor
**Presentation:** 1 poor
**Contribution:** 3 good
**Rating:** 5
**Confidence:** 1

**Summary:**

The authors study the problem of planning for efficient data collection. Given initial data with a lot of contexts but no reward, the question is how do you devise a policy for data collection such that when executed in the real world, it learns the reward optimally to finally learn a policy with maximum rewards.

They propose two methods for it, i) eluder planning and a sampling procedure
ii) uniform sampler for the case when the number of actions is small

**Strengths:**

Proposed algorithms for experiment planning and theoretically analyze them. They also showed suboptimality of the existing adaptive learning algorithms which is interesting.

Line 138-139, the intuition of the eluder dimension helps to understand its use case

**Weaknesses:**

The paper was hard for me to grab fully. Partially I lack background but majorly, I think it was the writing part. Some ideas to improve them:
1) Either go with a short, vague intro or a precise long intro: currently, it is a long but vague intro since the problem setting was not defined early, and it becomes difficult to understand. I have seen theoretical papers which define problem statements in the intro, and then it's a precise long intro.

2) Since you continuously refer to the sampling and planning phase, may be use a block diagram to represent them.

In general, I believe the writing style can be improved.

Secondly, I was wondering if it would make sense to also add at least a preliminary experiment showcasing your theory? It is hard to get intuition on how tight the bounds are.

**Questions:**

Intuitively when is it possible to have $d_{eluder}(F, B/T)$ a sublinear function in T?

I am not able to interpret theorem 4.1 properly, on what factors does $\tilde{c}$ depends. Consider writing it as: let T be the smallest integer satisfying the $T\geq....$, then \for all $T' \geq T$ $\pi_{T'}$ is optimal?

line 94,166: typos

**Limitations:**

no negative societal impact

---

> ### Author Rebuttal · Authors · 2023-08-08
>
> We deeply appreciate the reviewer’s comments. We will make sure the final version of our manuscript contains more detailed descriptions of the experiment planning setting early on in the introduction. We would like to remind the reviewer we are not the first to come up with this problem setting. The authors of [37] were the first to define the experiment planning problem. Our assumption is that drawing from the related literature while defining the problems we are studying is a valid choice. We will take the reviewer’s concern into account when writing the final version of this work and will make sure we carefully balance in our introduction the references to related work and what material to include in our own manuscript. We think the reviewer’s suggestion of including a block diagram showing the planning and sampling phases is a great idea. We will add a simple pictorial representation to the final version.
>
>
> Although we wholeheartedly agree with the reviewer that an experimental evaluation of these algorithms would be of great interest, we consider the main contributions of this work to be theoretical. As such, and due to the plethora of results we have, we think adding additional content would obfuscate the main contributions of the current manuscript. We believe that a thorough empirical exploration of the problem of experiment planning with function approximation would be an important contribution in itself that we would love to see our work followed up by. One important roadblock that such work will have to overcome is that in general optimistic algorithms may be intractable under function approximation for general function classes. This is an exciting area of future research.
>
> Answers to miscellaneous questions
>
>
> 1. “When is it possible to have $d_{\mathrm{Eluder}}(F, B/T)$ to be a sublinear function in $T$?. In order to guarantee learnability the eluder dimension needs to scale sublinearly in $T$. This is the case for plethora of function classes such as linear classes and generalized linear models (where $d_{\mathrm{Eluder}} (F, B/T) \approx d \log(B/T)$).
>
> 2. “I am not able to interpret theorem 4.1 properly, on what factors does $\tilde{c}$ depends”. We are somewhat confused at the reviewer’s comment here. The theorem statement specifies $\widetilde{c}$ is a universal constant (i.e. independent of the algorithms, or function classes). The result simply states, if one wants to produce an $\epsilon$-optimal policy $T \geq $[expression in the paper] samples are sufficient. The expression to the right of $T$ is dependent only on $\mathcal{F}, \mathcal{A}, \delta$ and $\epsilon$. We would greatly appreciate if the reviewer could further expand on what aspect of this formulation is confusing. We would love to hear these to improve our work.

---

> > ### Comment · Reviewer_ESHY · 2023-08-10
> > **about point 2**
> >
> > I was wondering, for intuition, when can you say $\tilde{c}$ will be large or small? Can it take any value e.g., $\infty$, if so why would be it a useful bound? or Maybe information about where does it arise would help me in understanding.

---

> > > ### Author Response · Authors · 2023-08-10
> > > **Constant $\tilde c$**
> > >
> > > Thank you so much for your question. The constant $\tilde c$ will be something like $100$ or so. This comes from a repeated application of Hoeffding bounds. Think of these constants as a substitute of $O$ or $\Omega$ notation. They are completely independent of $\mathcal{F}$, $\delta$ and any other problem dependent parameters. Instead of tracking the precise value of this constant (which would have made the computations tedious and burdensome) we instead substituted them by these symbols. We are happy to explain more if this is not clear.

---

> > > > ### Comment · Reviewer_ESHY · 2023-08-14
> > > > **Thanks for clarification**
> > > >
> > > > Thanks for the clarification. I have no more questions. I will keep my score.

---

### Official Review · Reviewer_qQSQ · 2023-07-12

**Soundness:** 2 fair
**Presentation:** 2 fair
**Contribution:** 2 fair
**Rating:** 5
**Confidence:** 5

**Summary:**

The study focuses on the experiment planning problem for contextual bandits with function approximation.
The paper gives two algorithms for the problem: (1) EluderPlanning algorithm whose sample complexity depends on the elder dimension of the function class and matches the sample complexity of OLS algorithm by a online-to-batch conversion.
(2) It also suggest that simple uniform sampling is good when action space is small, and the sample complexity matches the SquareCB algorithm by online-to-batch conversion.

Then, the paper suggest that there is an exponential gap between adaptive sampling and passive sampling (experimental planning) by finding a class of tree-structured function. Besides, for this class of function, the paper show that SquareCB and OLS (two adaptive sampling algorithm) have exponential large sample complexity while there exist an algorithm with reliability (called adaptive tree sampling in appendix) can achieve polynomial sample complexity in this tree-structured function class. This suggest the sub-optimality of SquareCB and OLS.

Finally, address the problem of model selection when the learner is presented with a family of reward function classes and the true reward function is inside one of the reward function classes.

**Strengths:**

The tree-structured function class constructed in section 5 is interesting and elegant.
With this function class, the paper point out an exponential separation between adaptive sampling algorithm.
Also, it points out that in realisable setting, the online-to-batch conversion of SquareCB and OLS is suboptimal for simple regret minimization.

**Weaknesses:**

(1) The proof of Lemma A.4 has a *major flaw*:
in Line 439, it is not valid to choose $\eta_t$ as a random variable that is dependent on the martingale difference sequence.
This is because the moment generating function (MGF) is required to derive lemma A.3 and thus also for lemma A.4.
It is valid only when the MGF coefficient $\eta$ is a fixed value, which is also reflected in the claim of lemma A.3 (eta is fixed outside the probability bound).
More generally, MGF coefficient $\eta$ can be a random variable that is independent of whole martingale difference sequence and then distribution of sum of martingale difference sequence is not affected by conditioning on $\eta$.

(2) Because of (1) and Lemma A.4 serves a fundamental step for most theoretical claims in appendix A, section 3 and section 6. All claims related to lemma A.4 may not be correct, including but not limited to Lemma A.5, Lemma A.7, Lemma A.8, Lemma 3.6, Theorem 3.4 and Proposition 6.1.

(3) There are a few typos that would affect the understanding of the technical details.

(4) Even if all claims can be corrected, the main contribution is not clean in the paper where too many issues are discussed.

**Questions:**

(1) See weaknesses. If all problems can be correctly resolved, I would consider increasing the score.

(2) There is a claim "Our results indicate the eluder dimension is not the sharpest statistical complexity measure to characterize learning in this setting." in line 326, but I don't find any support to this claim.

---

> ### Author Rebuttal · Authors · 2023-08-08
>
> We are extremely appreciative of the reviewer’s careful read of our manuscript. We are happy the reviewer identified the following strengths in our submission: this is the first work to explore the setting of experiment planning with function approximation and propose: a) Novel algorithms for Eluder Planning and Sampling and a thorough analysis of the Uniform Planner, B) the first lower bound drawing a gap between adaptive and static planning. We believe these contributions and in particular our lower bounds are of interest not only within the context of experiment planning but also more generally in the wider literature of bandits and reinforcement learning. The tree function class that forms the basis of our lower bound results shows how conditional structures embedded in adaptive learning problems are one of the components of what makes adaptive learning hard. The fact that existing adaptive learning algorithms based on for example the eluder dimension cannot give optimal rates in this example (as we show in this submission) implies the need for better more general algorithms for adaptive learning and a more complete treatment of the statistical complexity of this setting. We expect this insight to have wider consequences than what is contained in this submission.
>
> We will now address the reviewer’s main concerns. We apologize for the source of confusion and will try our best to explain it.
>
> We would like to start by noting the results of Lemma A.4 are true. An alternative proof of this results consists of modifying Theorem 2.2 from [“Bias no more: high probability data-dependent regret bounds for adversarial bandits and MDPs” ]. This result can be turned into an any-time bound by using a union bound over all $T \in \mathbb{N}$. Instead of applying the result with the probability level set to $\delta$ we use $\delta_T = \frac{\delta}{6*t^2}$. A union bound over all $T \in \mathbb{N}$ finalizes the result.
>
> The misunderstanding regarding the correctness of this lemma is the result of us authors deciding to add a proof based on a support lemma (Lemma A.3) borrowed from Lemma 9 in the paper entitled “Taming the Monster: A fast and simple algorithm for contextual bandits”. In the notation from the Taming the monster paper, it was our understanding that this result was true uniformly over all lambda values (our eta) for any fixed delta. This is how the authors of the “Taming the monster” paper seem to use it in the proof of their Lemma 11.
>
> Due to the reviewer’s careful reading of our manuscript we dug deeper into the origins of Lemma A.3 (Lemma 9 in Taming the monster”) and discovered that in its original form (Beygelzimer et al. (2011) ), the result holds for a specific delta and gamma. Although this seems to imply the taming the monster paper has a slight error their results are still true. Had Lemma 9 in “taming the monster” held uniformly - as we initially thought-, we thought setting the eta value to a random variable to be well justified. As we have mentioned before there are many ways to fix this issue. As we have mentioned above in our case one way is to instead derive Lemma A.4 using a uniform version of Theorem 2.2 from “Bias no more”. An alternative proof avenue is by invoking the results from Howard [2018] “Time-uniform, nonparametric, nonasymptotic confidence sequences”, although using these methods require more technical jargon in the form of sub-psi processes that we wanted to avoid using in our manuscript. In the case of the “Taming the monster results”, we believe it is also possible to derive a uniform in lambda (our eta) version of Lemma 9 from that paper (our Lemma A.3) by taking a union bound over an exponential grid of the lambda space. It may be of interest to the community to make the authors of that work aware of this issue.
>
> We again want to thank the reviewer for catching this and would really appreciate that in case this explanation has been found to be sufficient, if the reviewer could consider the technical merits of the remainder of the paper and raise their score if they think it to be appropriate.
>
> The reviewer mentions the confusing nature of the claim “our results indicate eluder is not the sharpest statistical complexity measure to characterize learning in this setting”. In this case we were simply referring to the gap that exists between the eluder planner and sampler algorithms and the uniform sampling strategy. When the size of the action set is very small the uniform sampling strategy will yield a sharper upper bound than the eluder bound. A really exciting area of future research is to understand the tradeoffs that exist between these two settings and figuring whether there is a single algorithmic solution and a more nuanced notion of statistical complexity that can bridge the gap between them.

---

> > ### Comment · Reviewer_qQSQ · 2023-08-10
> > **About the major concern**
> >
> > If you are going to use Theorem 2.2 in "Bias no more" to replace the lemma A.4 in your paper, you might need to justify all of your result remain the same at least in the dominant term. It might need a huge amount of derivations from the beginning, which might be another paper then.

---

> > > ### Author Response · Authors · 2023-08-10
> > > **The two lemmas are equivalent**
> > >
> > > Dear reviewer we are confused by this question as Lemma A.4 and the anytime version we described above of Lemma 2.2 in "Bias no more" are exactly the same up to constants -and log factors.
> > >
> > > $V$ in bias no more is exactly $\sum_{\ell=1}^T E_\ell[X_\ell^2]$ of our submission. And $B^*$ in bias no more is exactly $R$ in ours.
> > >
> > > We are happy to explain more why it is exactly the same up to constants and log factors if this explanation is not clear.
> > >
> > > The any-time version of Lemma 4 in "Online Model Selection for Reinforcement Learning with Function Approximation" can also serve as a proof of Lemma A.4. This version would not suffer any logarithmic blowup. We are happy to provide an explanation if it is not clear.

---

> > > > ### Comment · Reviewer_qQSQ · 2023-08-10
> > > > **Request for more details**
> > > >
> > > > Sure, please explain more starting from theorem 2.2 in "bias no more". Thank you!

---

> > > > > ### Author Response · Authors · 2023-08-10
> > > > > **Step by step explanation**
> > > > >
> > > > > Theorem 2.2 in Bias no More states that for any $\delta_T \in (0,1)$
> > > > >
> > > > > $\sum_{t=1}^T X_t \leq C \sqrt{ V \ln(C/\delta_T) } + 2CB\ln(C/\delta_T)  $ (1)
> > > > >
> > > > > with probability at least $1-\delta_T$ where $V = \sum_{t=1}^T E_t[X_t^2]$ and $B = \max{ | X_i|} \leq R$ and $C = \log^2(B) + \log(B)\log(T)$. The anytime version of this lemma is
> > > > >
> > > > > $\sum_{t=1}^T X_t \leq C \sqrt{ V \ln(6CT^2/\delta) } + 2CB\ln(6T^2C/\delta)  $ (2)
> > > > >
> > > > > for all $T \in \mathbb{N}$ with probability at least $1-\delta$. This can be proven by applying (1) setting $\delta_T = \delta/6T^2$, a union bound and using the fact that $\sum_{t=1}^\infty \frac{1}{t^2} < 6$.
> > > > >
> > > > > Substiting the definitions of $V$ and $B$ into equation (2) yields,
> > > > >
> > > > > $\sum_{t=1}^T X_t \leq C' (\log^2(R )+ \log(R)\log(T))\sqrt{ \sum_{t=1}^T E_t[X_t^2] \ln(T*(\log^2(R )+ \log(R)\log(T)))/\delta)} + C' (\log^2(R )+ \log(R)\log(T))R\ln(T(\log^2(R )+ \log(R)\log(T)))/\delta) $
> > > > >
> > > > > For some \textbf{universal constant} $C'$. This is because trivially $\ln(6CT^2/\delta)  = \mathcal{O}( \ln( T*(\log^2(R )+ \log(R)\log(T)))/\delta  ) )$.
> > > > >
> > > > >
> > > > > This recovers Lemma A.4 up to logarithmic factors (i.e. if we call $C = (\log^2(R )+ \log(R)\log(T)))$) the bound above looks like,
> > > > >
> > > > > $\sum_{t=1}^T X_t \leq C'C\sqrt{ \sum_{t=1}^T E_t[X_t^2] \ln(CT/\delta)} + C' C R\ln(CT/\delta) $.
> > > > >
> > > > > with probability at least $1-\delta$ for all $T \in \mathbb{N}$.

---

> > > > > > ### Author Response · Authors · 2023-08-14
> > > > > > **More Questions?**
> > > > > >
> > > > > > Dear Reviewer,
> > > > > >
> > > > > > We would like to know if this explanation made sense? If so and this addresses the reviewer's concerns about correctness, we would very much appreciate if the reviewer could reassess their score to one that reflects this renewed understanding.
> > > > > >
> > > > > > Thanks a lot!

---

> > > > > > > ### Comment · Reviewer_qQSQ · 2023-08-14
> > > > > > > **Reply to the comments**
> > > > > > >
> > > > > > > Thank you for your explanation. I would consider reassessing the score.

---

> > > > > > > > ### Author Response · Authors · 2023-08-18
> > > > > > > > **Thanks**
> > > > > > > >
> > > > > > > > We are really grateful for the reviewer's comments and time. This discussion will definitely help improve the final version of this work.

---

### Official Review · Reviewer_aXBj · 2023-07-14

**Soundness:** 3 good
**Presentation:** 3 good
**Contribution:** 3 good
**Rating:** 6
**Confidence:** 3

**Summary:**

We study the static experiment planning for policy learning problem in contextual bandits, with focus on the general realizable case. The paper first presents an algorithm using reward free, extending similar ideas in [37] and leveraging the  eluder dimension. The paper then shares a few theoretical results, including the competitive guarantee for uniform sampling policy, the gap between static and adaptive policies in a special case, and the results in model selection.


**Strengths:**

* The paper is technically sound.
* The writing is very clear.
* The paper studies an important problem and the first proposed algorithm is useful.


**Weaknesses:**

Numerical results on the finite sample performance would be important to have. For example, although the asymptotic rate for uniform sampling is competitive and this fact might be surprising, it is expected that its finite-sample performance is worse.

It is also not fully clear how significant these theoretical findings are. For example, it is expected that there is some performance gap between adaptive and static policies, when the task is hard (though I appreciate formally proving it).



**Questions:**

1. It would be helpful to add outputs to your Algorithm tables
2. In the main text, it would help to discuss the connection to [37] in terms of the reward-free emulation idea.
3. "Extracting Policy from Data" - the intuition for this part is hard to understand. Why uniform sampling + optimistic, instead of just using the greedy (or pessimistic) policy?


**Limitations:**

Extensions to safe exploration and/or policy evaluation tasks would be meaningful and worthy some discussions.

---

> ### Author Rebuttal · Authors · 2023-08-08
>
> “Numerical results on the finite sample performance would be important to have” Although we wholeheartedly agree with the reviewer that an experimental evaluation of these algorithms would be of great interest, we consider the main contributions of this work to be theoretical. As such, and due to the plethora of results we have, we think adding additional content would obfuscate the main contributions of the current manuscript. We believe that a thorough empirical exploration of the problem of experiment planning with function approximation would be an important contribution in itself that we would love to see our work followed up by. One important roadblock that such work will have to overcome is that in general optimistic algorithms may be intractable under function approximation for general function classes. This is an exciting area of future research.
>
> The reviewer raises a couple of good clarification points. If our work gets accepted at the conference we will make use of the extra space to better explain a few things. We will add a couple of explanations here to hopefully allay the reviewer’s concerns, we will add richer explanations in the final version of our manuscript.
>
> 1. “It is also not fully clear how significant these theoretical findings are.” this is the first work to explore the setting of experiment planning with function approximation. We believe our manuscript has a couple of important results, for example a) Novel algorithms for Eluder Planning and Sampling and a thorough analysis of the Uniform Planner, B) the first lower bound drawing a gap between adaptive and static planning. We believe these contributions and in particular our lower bounds are of interest not only within the context of experiment planning but also more generally in the wider literature of bandits and reinforcement learning.
>
> 2. “it would help to discuss the connection to [37] “ Our work subsumes that of [37] since in the linear case our Eluder Planner and Sampler procedures recover the same rates as [37]. We will make sure this is fully explained in the final version of our work.
>
> 3. “Extracting Policy $\hat{\pi}_T$ from Data - the intuition for this part is hard to understand”. We prove the Eluder Sampler Planner procedure works by showing the simulated $\widetilde \pi_t^{\mathrm{opt}}$ sequence (produced via optimistic evaluations) satisfies a regret bound. This implies that sampling a uniform policy can be used to turn this ‘online’ regret bound to a PAC guarantee. The same argument would not work under a greedy or pessimistic policy. This does not preclude it being possible to show either greedy or pessimistic policies would be appropriate choices, but it would require a whole set of different techniques to prove this. We agree this is an interesting follow up research question.
>
> 4. “Extensions to safe exploration and/or policy evaluation tasks would be meaningful” The reviewer has identified another set of interesting follow up directions. The problem of experiment planning under constraints is an important and yet unexplored area of research. We would be really happy to see more research efforts being spent towards better understanding all facets of experiment planning.

---

> > ### Author Response · Authors · 2023-08-18
> > **Follow up**
> >
> > Dear Reviewer,
> >
> > We wanted to reiterate our commitment to have the reviewer's questions addressed. Please let us know if the response above addressed the reviewer's concerns. If these have been adequately addressed we would very much appreciate any indication of this.
> >
> > Thanks so much!
> >
> > The Authors

---

> > > ### Comment · Area_Chair_V2Qq · 2023-08-18
> > >
> > > Dear authors,
> > >
> > > Please do not ask the reviewers to increase score.
> > >
> > > Your AC

---

> > > > ### Comment · Reviewer_aXBj · 2023-08-21
> > > > **Thank you for the rebuttal**
> > > >
> > > > I read the rebuttal and my opinion has not changed. In fact, I believe adding numerical results and lower bounds would improve the paper a lot, and at least the numerical study should not be left to follow-up work.

---

### Decision · Program_Chairs · 2023-09-21

**Decision:**

Accept (poster)

**Comment:**

This paper studies experiment design with a general function approximation in the bandit setting. The design is a sequence of $T$ data logging policies, which are computed offline and then executed online. After I read all reviews, my overall perception of the paper was:

* **Theory:** The paper is quite theoretical and not easy to read. Intuitions for the proofs are missing.

* **Experiments:** None. While I understand that this is a theory paper, simple synthetic experiments would convince the reader that this paper can have impact on practice.

* **Lower bounds:** None. How else can we tell that the proposed solution is efficient?

On the top of this, the authors responded that lower bounds are hard and experiments are not needed. But how can we tell then that the algorithm design in this work is not highly suboptimal? To decide, I read the paper and my summary is below:

* **Section 3:** Eluder dimension with a known function class is used to design an experiment. A $T$-round static policy is computed in Algorithm 1 and executed in Algorithm 2. In Theorem 3.4, a well-known cumulative regret to simple regret reduction is used to get simple regret guarantees.

* **Section 4:** The same as Section 3 but with uniform sampling. At the end of the section, the authors argue that the proposed $T$-round policies are good, in the sense that a trivial cumulative regret to simple regret reduction applied to OLS and SquareUCB has the same guarantees.

* **Section 5:** A trivial cumulative regret to simple regret reduction applied to OLS and SquareUCB can be suboptimal.

* **Section 6:** Generalization to multiple function classes.

The main contribution of this work are non-adaptive policies with general function classes that are comparable to the adaptive ones, with a trivial cumulative regret to simple regret reduction. There are some obvious shortcomings:

* No comparison to any adaptive simple regret minimization algorithm from the best-arm identification (BAI) literature. This line of works started with [Best Arm Identification in Multi-Armed Bandits](http://certis.enpc.fr/~audibert/Mes%20articles/COLT10.pdf). The optimal rates in BAI are $\exp[- T]$ and there are also well-established lower bounds. Essentially all results in this work rely on a trivial cumulative regret to simple regret reduction, which is not optimal.

* Algorithm 1 is not computationally efficient unless the function class is simple (a finite set of known functions).

* The eluder dimension may not be the right metric. The reason is that adaptivity to rewards is clearly useful beyond linear models because their confidence intervals depend on the rewards, as in logistic models.

The authors also missed a few related works:

* [Safe Optimal Design with Applications in Off-Policy Learning](https://proceedings.mlr.press/v151/zhu22a.html)

* [Safe Exploration for Efficient Policy Evaluation and Comparison](https://proceedings.mlr.press/v162/wan22b.html)

Despite all shortcomings, I enjoyed reading this paper and support its acceptance. Please include all my suggestions in the next version of the paper.